# Why Specialist Models Still Matter: A Heterogeneous Multi-Agent Paradigm for Medical Artificial Intelligence

Yanan Wang [* 1]  Shuaicong Hu [* 1]  Jian Liu [1]  Guohui Zhou [1]  Aiguo Wang [2]  Cuiwei Yang [1]

## Abstract

The impressive performance of generalist large language models (LLMs) such as GPT and Claude in healthcare raises a critical question: will domain-specific medical specialist models become obsolete? We argue that the future of medical artificial intelligence (AI) lies not in building monolithic medical foundation models, nor in replacing human expertise, but in orchestrating collaboration among generalist LLMs, domain-specific specialist models, and clinicians. We propose HetMedAgent, a heterogeneous medical multi-agent framework that enables conflict-aware evidence fusion, uncertainty-based clinician intervention triggering, and adaptive threshold calibration. Experiments on three real-world clinical decision-making tasks demonstrate that the synergy between generalist LLMs and domain-specific specialist models significantly outperforms using either type of model alone, validating the irreplaceable value of specialist models in modality-specific analysis. HetMedAgent represents a shift from building medical LLMs or foundation models to multi-agent collaboration, achieving a balance between general reasoning capabilities and domain-specific precision.

## 1 Introduction

The rapid advancement of large language models (LLMs) has sparked intense debate about their role in healthcare (Thirunavukarasu et al., 2023; Lee et al., 2023). As generalist LLMs like GPT(Achiam et al., 2023), Claude (Anthropic AI, 2024), and DeepSeek (Guo et al., 2025) demonstrate impressive capabilities in medical question-answering (Singhal et al., 2023; Nori et al., 2023) and clinical reasoning (Liévin et al., 2024; Saab et al., 2024), a critical question emerges: Are domain-specific medical specialist models becoming obsolete? While tempting, this view overlooks the fundamental complexities and constraints unique to the healthcare domain.

We advocate for a more nuanced perspective. The development of medical foundation models faces substantial obstacles that generalist LLMs need not directly confront, yet cannot truly circumvent in medical applications. Medical data is inherently scarce, fragmented across institutions, and bound by strict privacy regulations (Cohen & Mello, 2018; Voigt & Von dem Bussche, 2017). More critically, deploying AI in healthcare involves profound ethical concerns—incorrect decisions can be life-threatening, and accountability remains unclear (Gerke et al., 2020; Reddy et al., 2021). These challenges suggest that rather than abandoning domain-specific medical specialist models, we should orchestrate collaboration between generalist LLMs and domain-specific specialist models, leveraging the reasoning capabilities of the former and the precision of the latter. Second, and equally important, our goal is not to replace clinicians. Despite AI's growing capabilities, physicians bring irreplaceable elements to medical decision-making: years of clinical experience, nuanced judgment in ambiguous cases, ethical reasoning, and the capacity to handle the human dimensions of healthcare (Topol, 2019; Jabbour et al., 2023; Takita et al., 2025). In our framework, clinicians themselves are agents, playing critical roles in final decision-making, handling edge cases, and ensuring patient safety.

This motivates our proposed heterogeneous medical multi-agent collaborative framework, HetMedAgent, where three types of agents—generalist LLMs, specialist models, and clinicians—constitute an indispensable architecture for medical decision-making (Figure 1). Generalist LLMs serve as coordinators and reasoning engines, capable of understanding complex clinical narratives, decomposing tasks, and integrating heterogeneous information (Wei et al., 2022; Yao et al., 2023; Hao et al., 2025). Domain-specific specialist models, trained for tasks such as medical image interpretation and physiological signal analysis,

---
[*]Equal contribution   [1]Department of Biomedical Engineering, College of Biomedical Engineering, Fudan University, Shanghai 200433, China [2]Department of Cardiology, Xinghua City People's Hospital affiliated to Yangzhou University, Jiangsu 225700, China. Correspondence to: Guohui Zhou <zhough@fudan.edu.cn>, Aiguo Wang <xhwag@yzu.edu.cn>, Cuiwei Yang <yangcw@fudan.edu.cn>.

*Proceedings of the 43rd International Conference on Machine Learning*, Seoul, South Korea. PMLR 306, 2026. Copyright 2026 by the author(s).

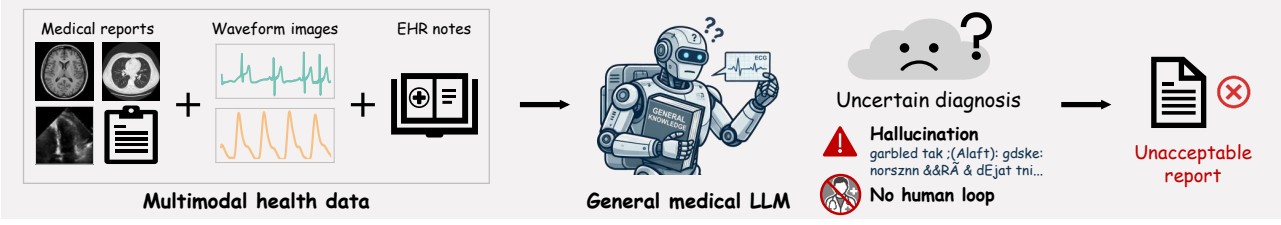

**a** Previous: The black-box generalist

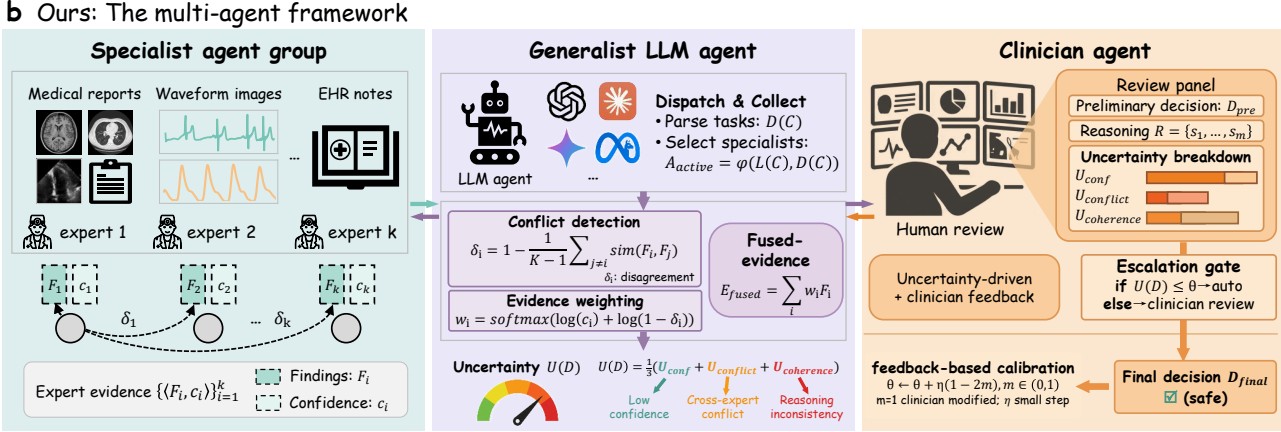

**b** Ours: The multi-agent framework

*Figure 1.* (a) Previous: The generalist medical LLM. Lacking domain specificity and human oversight, single LLMs are prone to hallucinations and unsafe decisions on multimodal data. (b) Ours: Overview of the HetMedAgent framework. Orchestrates a Specialist Agent Group, a Generalist LLM Agent, and a Clinician Agent, enabling conflict-aware evidence fusion and uncertainty-driven oversight with adaptive calibration.

act as domain experts, providing deep, focused analysis in their respective areas (van Leeuwen et al., 2021; Hong et al., 2024; Hu et al., 2025). Clinicians serve as final arbiters, ethical gatekeepers, and handlers of cases that exceed AI capabilities or carry high stakes (Jacobs et al., 2021).

This framework represents a fundamental shift in medical AI (Xiong et al., 2023), advocating collaborative intelligence where each agent contributes unique strengths. Key advantages include: (1) leveraging existing generalist LLMs without expensive medical-specific pretraining; (2) maintaining specialist model precision in narrow domains; (3) ensuring accountability through human oversight; (4) supporting modular incorporation of new specialized models; (5) enhancing interpretability via explicit collaboration chains (Ghassemi et al., 2021). Our experimental results validate this approach. Generalist LLMs collaborating with domain-specific specialist models effectively serve clinical decision-making across admission risk stratification, etiology prediction, and disease severity assessment. HetMedAgent achieves performance comparable to or exceeding medical foundation models (Singhal et al., 2023; Xiong et al., 2023) with significantly lower development costs. By integrating clinicians as active decision-loop participants, we enhance accuracy, physician trust, and preserve the indispensable human element in healthcare (Yang et al., 2022; Tonekaboni et al., 2019).

**Our key contributions are summarized as follows:**

- **Heterogeneous Multi-Agent Architecture:** We propose a principled framework that orchestrates collaboration among generalist LLMs, domain-specific specialist models, and human clinicians, enabling each agent type to contribute its unique strengths while compensating for individual limitations.

- **Uncertainty-Based Routing:** We introduce a multi-dimensional uncertainty quantification mechanism that dynamically assesses confidence across modality-specific analyses, cross-agent agreement, and reasoning chain coherence, enabling intelligent decision routing and clinician intervention triggering.

- **Dynamic Collaboration Protocol:** We design a structured interaction protocol that enables conflict-aware evidence fusion, uncertainty-based escalation to clinician oversight, and adaptive threshold calibration based on clinician agent feedback, ensuring both efficiency and safety in clinical decision-making.

## 2 Related Works

### 2.1 Large Language Models in Healthcare

Generalist LLMs show promise in medical examinations and clinical reasoning (Nori et al., 2023; Singhal et al., 2023), yet exhibit insufficient domain knowledge, hallucinations, and limited understanding of complex cases

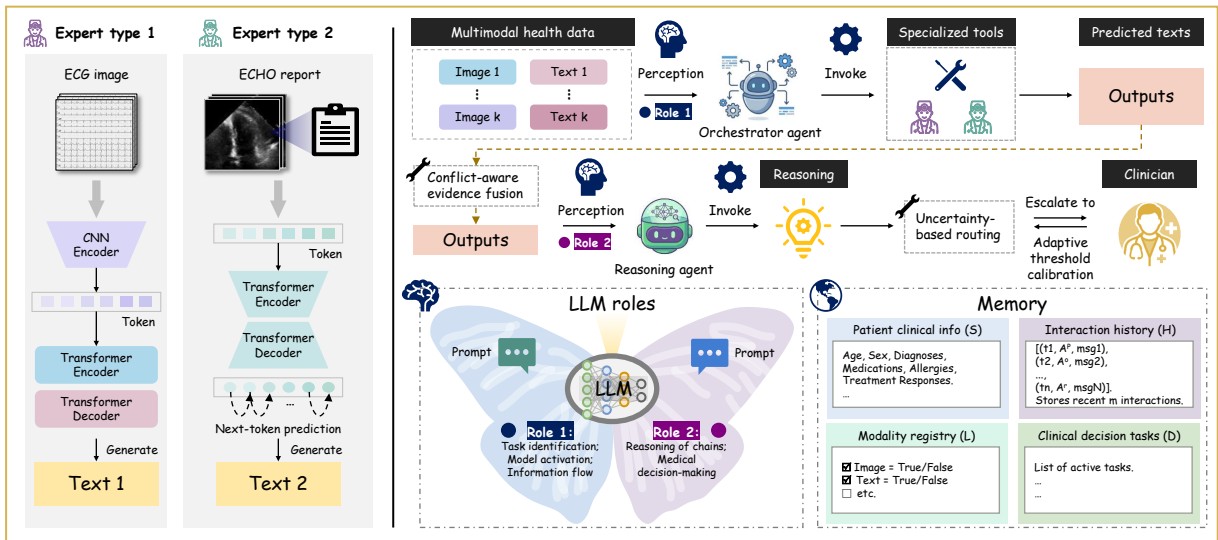

*Figure 2.* Detailed architecture of the HetMedAgent system. (**Left**) Specialist models convert multimodal data into standardized text. (**Top-Right**) Orchestration and reasoning workflow. (**Bottom-Right**) Shared memory module.

(Thirunavukarasu et al., 2023; Lee et al., 2023). Medical LLMs like Med-PaLM (Singhal et al., 2023) improve through fine-tuning on medical data (Saab et al., 2024; Yang et al., 2022) but face high training costs and multimodal performance gaps (Rajpurkar et al., 2022). Existing work treats models as isolated tools (Liévin et al., 2024) without exploring synergistic integration of generalist reasoning and specialist precision, motivating our HetMedAgent that coordinates generalist LLMs, specialist models, and clinicians.

## 2.2 Medical Foundation Models and Specialist Models

Medical foundation models pre-trained on large-scale medical data provide domain representations, including PubMedBERT (Gu et al., 2021), BioBERT (Lee et al., 2020), and BiomedCLIP (Zhang et al., 2023). Domain-specific models for tasks like diabetic retinopathy screening (Gulshan et al., 2016), skin cancer classification (Esteva et al., 2017), and atrial fibrillation detection (Hannun et al., 2019) achieve clinician-level performance using ResNet (He et al., 2016) and EfficientNet (Tan & Le, 2019), but lack cross-domain reasoning.

## 2.3 Multi-Agent Systems

Multi-agent systems coordinate autonomous agents for complex tasks (Dorri et al., 2018). LLM-based frameworks like AutoGen (Wu et al., 2023) and MetaGPT (Hong et al., 2024) enable natural language interaction. However, medical applications remain limited (Reddy et al., 2021). Agent-Clinic (Schmidgall et al., 2024) uses multiple GPT-4 instances as specialists but lacks domain-specific models and clinician integration. Our HetMedAgent integrates generalist reasoning, specialist precision, and clinical experience.

## 2.4 Human-AI Collaboration in Healthcare

Human-AI collaboration is critical for safe medical AI deployment. Challenges include automation bias where physicians over-rely on AI (Jacobs et al., 2021), and lack of explainability undermining trust (Ghassemi et al., 2021; Takahashi et al., 2025). Multi-agent collaboration with clinicians remains underexplored regarding intervention triggering and interaction protocols (Tonekaboni et al., 2019; Shimgekar et al., 2025). HetMedAgent positions physicians as core agents with final authority, integrating uncertainty-based triggering and structured protocols.

## 2.5 Team-Based Decision-Making and MDT Workflows

Clinical decision-making has long been collaborative. Multidisciplinary Team (MDT) workflows such as tumor boards separate evidence gathering (modality specialists) from decision synthesis (attending physician) (Lamb et al., 2011). HetMedAgent mirrors this paradigm: specialist agents gather modality-specific evidence while the orchestrator synthesizes recommendations. Recent work on human-AI teaming supports this design—structured delegation outperforms single generalist decision-makers (Munyaka et al., 2023), and joint decision quality improves when algorithmic output is matched to user expertise (Inkpen et al., 2023), aligning with our uncertainty-based routing that escalates cases when system confidence is insufficient.

## 3 Methodology

Figure 2 illustrates the overall architecture of HetMedAgent. The framework takes patient cases with multimodal medical data as input and leverages an orchestrator agent to coordinate specialist agents for modality-specific analysis. The orchestrator integrates specialist outputs through

a reasoning agent, quantifies decision uncertainty, and triggers clinician intervention when needed. In the following sections, Section 3.1 formally defines the problem, Section 3.2 details the heterogeneous agent architecture, and Section 3.3 describes the dynamic collaboration protocol. Notations are summarized in Appendix A.

## 3.1 Problem Definition

Given a patient case $C = \{V, I\}$, where $V$ denotes clinical information (age, gender, chronic disease history, treatment history, symptoms) and $I$ comprises multimodal examination data (imaging, physiological signals, etc.), our goal is to generate clinical decisions $D = \{d_1, \ldots, d_k\}$ that maximize decision quality $Q(D, C)$ while satisfying safety and efficiency constraints.

Traditional single-model approaches $D = M(C)$ struggle with multimodal cases, as no single model masters all specialist domains. Our HetMedAgent decomposes decision-making into subtasks collaboratively solved by heterogeneous agents:

$$D = A^P \left( A^r \left( \{A_i^w\}(\mathcal{O}(C)) \mid V, I \right) \right) \quad (1)$$

where orchestrator $\mathcal{O}$ dispatches modality-specific data $I_i$ to specialist agents $A_i^w$, producing $F_i^w$ with finding $F_i$ and generation confidence $c_i \in [0, 1]$. The system computes integration weights $w_i$. Reasoning agent $A^r$ integrates $\{F_i^w, w_i\}$ with clinical information $V$ to generate preliminary decisions, which are reviewed by clinician agent $A^P$ when uncertainty exceeds a threshold, producing final decisions $D$.

## 3.2 HetMedAgent Architecture

Our HetMedAgent comprises: Memory module for information storage; orchestrator $\mathcal{O}$ for task coordination; specialist agents $A_i^w$ for modality-specific analysis; reasoning agent $A^r$ for integrated decision-making; and clinician agent $A^P$ for supervision and parameter optimization. Additionally, the framework includes two key functional modules: conflict-aware evidence fusion for integrating specialist outputs with dynamic weighting, and uncertainty-based routing for intelligent decision escalation to clinicians.

### 3.2.1 MEMORY MODULE

The memory module stores patient clinical information $V(C)$, interaction history $H(C)$, modality registry $L(C)$ for dynamic agent activation, and clinical tasks $D(C)$. The orchestrator constructs comprehensive context $\text{Context}(C) = \{V(C), H(C), L(C), D(C)\}$ for all agents.

### 3.2.2 ORCHESTRATOR AGENT

The orchestrator $\mathcal{O}$, built on generalist LLMs, serves as the central coordinator with three core responsibilities: (1) parsing clinical decision-making tasks from input; (2) dynamically activating specialist agents based on available

modalities; (3) managing inter-agent information flow and execution order.

**Task Identification.** Upon receiving case $C$, $\mathcal{O}$ retrieves context $\text{Context}(C)$ from memory and identifies tasks via structured prompt $\psi_{\text{task}}$ (Figure 8 in Appendix B):

$$D(C) = \text{LLM}_{\mathcal{O}}(\psi_{\text{task}}, \text{Context}(C)) \quad (2)$$

where $\psi_{\text{task}}$ guides the model to identify decision types. For complex cases, $\mathcal{O}$ decomposes tasks into subtasks, each targeting specific modalities or clinical questions.

**Agent Activation.** Based on modality registry $L(C)$ and identified tasks $D(C)$, $\mathcal{O}$ determines which specialist agents to activate via mapping function $\varphi$:

$$A_{\text{active}} = \varphi(L(C), D(C)) \quad (3)$$

where $\varphi : L \times D \to 2^{A^w}$ maps modalities and tasks to specialist agent subsets. For instance, when $L(C)$ contains electrocardiogram (ECG) data and $D(C)$ involves cardiovascular risk, $\varphi$ activates ECG specialist agent $A_{\text{ECG}}^w$.

**Execution Coordination.** $\mathcal{O}$ dispatches task instructions to activated specialists using parallel execution for efficiency. After all specialist analyses complete, $\mathcal{O}$ triggers reasoning agent $A^r$ for multimodal fusion. All interactions are logged to $H(C)$ for traceability.

### 3.2.3 SPECIALIST AGENTS

Each specialist agent $A_i^w$ focuses on a specific modality and outputs standardized findings with confidence scores:

$$F_i^w = \{\text{diagnosis} : F_i, \text{confidence} : c_i \in [0, 1]\} \quad (4)$$

where $c_i$ is the generation confidence, computed as the geometric mean of per-token softmax probabilities (equivalently, inverse perplexity).

Specialist architectures are modality-dependent, using Transformer encoder-decoders for text inputs and CNN encoders with Transformer decoders for images. The framework design allows dynamic addition of new specialist agents without modifying core architecture. New specialists only need to implement the standard modular interface, enabling the system to evolve with advances in medical AI.

### 3.2.4 CONFLICT-AWARE EVIDENCE FUSION

**Conflict Detection.** Each specialist finding $F_i$ is projected into a semantic embedding space using a PubMedBERT-based bi-encoder with shared-direction adjustment to mitigate phrasing biases (details in Appendix C), yielding embeddings $\tilde{\mathbf{e}}_i$. For each specialist $i$, the conflict score $\delta_i$ measuring its disagreement with other specialists is:

$$\delta_i = 1 - \frac{1}{k-1} \sum_{j \neq i} \frac{1 + \text{sim}(\tilde{\mathbf{e}}_i, \tilde{\mathbf{e}}_j)}{2} \quad (5)$$

where $\text{sim}(\cdot, \cdot)$ denotes cosine similarity and $k$ is the total number of specialists. This captures semantic inconsis-

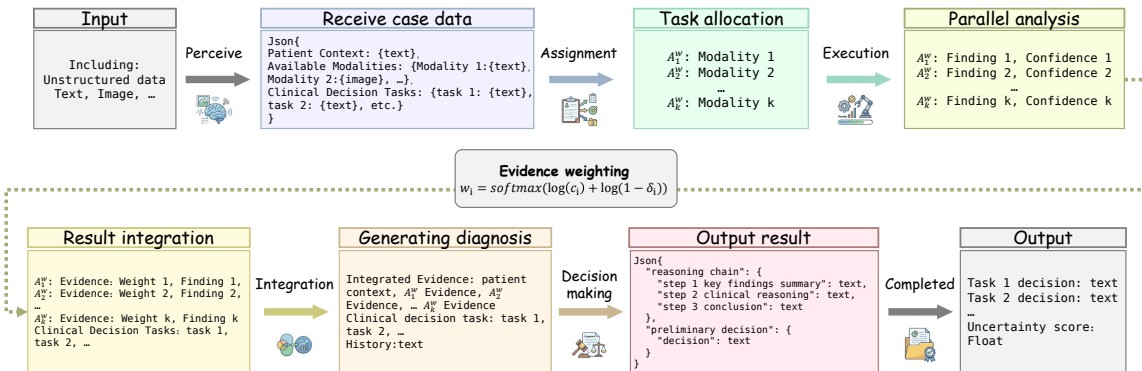

*Figure 3.* HetMedAgent's complete decision-making workflow with inputs and outputs.

tency rather than lexical overlap.

**Weighted Evidence Fusion.** The system receives outputs $\{F_1^w, F_2^w, \ldots, F_k^w\}$ from all specialist agents and computes integration weights to resolve conflicts. We assemble the weighted evidence into a single reasoning input via a deterministic evidence assembly protocol:

$$\text{Input}_{\text{reason}} = \text{Context}(C) \oplus \bigoplus_{i=1}^{k} \langle F_i^w, w_i \rangle \quad (6)$$

where $\oplus$ denotes string concatenation. Each $\langle F_i^w, w_i \rangle$ pair is presented as a structured annotation in the prompt to the reasoning agent, guiding it to attend more to relatively reliable specialist evidence. For each case, the specialist-specific weights $w_i$ are computed from the generation confidence $c_i$ and conflict score $\delta_i$:

$$w_i = \text{softmax}(\log(c_i) + \log(1 - \delta_i)) \quad (7)$$

Instead of using a product-of-experts fusion over $w_i$, we use $w_i$ as prompt-level evidence annotations because product-of-experts would require calibrated probability distributions over a shared label space, which is not available for natural-language findings across complementary modalities.

### 3.2.5 REASONING AGENT

The reasoning agent $A^r$ is the key component for generating preliminary decisions. Built on generalist LLMs (can use the same or different model as orchestrator $\mathcal{O}$), $A^r$ leverages powerful reasoning and knowledge integration capabilities to generate coherent clinical decisions. $A^r$ receives the assembled evidence input $\text{Input}_{\text{reason}}$ (Eq. 6) and, guided by structured prompt $\psi_{\text{reason}}$ (Figure 9 in Appendix B), generates preliminary decision $D_{\text{prelim}}$:

$$D_{\text{prelim}} = \text{LLM}_{A^r}(\psi_{\text{reason}}, \text{Input}_{\text{reason}}) \quad (8)$$

The prompt $\psi_{\text{reason}}$ instructs the model to: (1) make decisions with clinical knowledge and medical guidelines; (2) explicitly state reasoning chains and key evidence.

**Reasoning Chain Generation.** To enhance interpretability, $A^r$ explicitly generates reasoning chains $\mathcal{R} =$

$\{s_1, s_2, \ldots, s_m\}$ capturing step-by-step clinical logic. The coherence score measures reasoning chain consistency via inter-step semantic similarity:

$$U_{\text{coherence}} = 1 - \frac{1}{m-1} \sum_{t=1}^{m-1} \text{sim}(s_t, s_{t+1}) \quad (9)$$

where $\text{sim}(\cdot, \cdot)$ denotes cosine similarity between step embeddings. Lower scores indicate more consistent logical flow, helping clinicians understand AI decision processes and providing traceable evidence for audit and quality control.

### 3.2.6 UNCERTAINTY-BASED ROUTING

**Uncertainty Quantification.** While generating preliminary decision, the system computes comprehensive uncertainty score $U(D_{\text{prelim}})$ to trigger clinician intervention. Uncertainty comprises three dimensions:

$$U(D_{\text{prelim}}) = \lambda_{\text{conf}} U_{\text{conf}} + \lambda_{\text{conflict}} U_{\text{conflict}} + \lambda_{\text{coherence}} U_{\text{coherence}} \quad (10)$$

where: $U_{\text{conf}} = 1 - \max_i(c_i)$ reflects confidence gaps; $U_{\text{conflict}} = \frac{1}{k} \sum_i \delta_i$ measures mean specialist disagreement; $U_{\text{coherence}}$ quantifies reasoning chain incoherence from the generated reasoning process. The coefficients $\lambda_{\text{conf}}, \lambda_{\text{conflict}}$, and $\lambda_{\text{coherence}}$ are non-negative contribution weights satisfying $\lambda_{\text{conf}} + \lambda_{\text{conflict}} + \lambda_{\text{coherence}} = 1$. In our three-component experimental setting, we assign equal contribution to all components, i.e., $\lambda_{\text{conf}} = \lambda_{\text{conflict}} = \lambda_{\text{coherence}} = \frac{1}{3}$; when a component is structurally absent or ablated, weights are re-normalised over the remaining applicable components.

**Threshold-Based Decision.** Based on uncertainty score $U(D_{\text{prelim}})$ and preset threshold $\theta_P$, the system decides whether to trigger clinician intervention:

$$D_{\text{output}} = \begin{cases} D_{\text{prelim}}, & \text{if } U(D_{\text{prelim}}) \leq \theta_P \\ A^P(D_{\text{prelim}}, \mathcal{R}), & \text{if } U(D_{\text{prelim}}) > \theta_P \end{cases} \quad (11)$$

When $U(D_{\text{prelim}}) > \theta_P$, the system escalates to clinician agent $A^P$ for review. Crucially, HetMedAgent operates as a Clinical Decision Support System (CDSS): even in "au-

*Table 1.* Main results: HetMedAgent vs. medical LLMs and multi-agent systems on cardiovascular clinical decision-making tasks. Best results in **bold**.

| Method | Risk Stratification | | Etiology | | Severity | |
|---|---|---|---|---|---|---|
| | AUROC ↑ | F1 ↑ | AUROC ↑ | F1 ↑ | AUROC ↑ | F1 ↑ |
| *Medical LLMs* | | | | | | |
| PMC-LLaMA (Wu et al., 2024) | 0.782 | 0.745 | 0.698 | 0.652 | 0.641 | 0.598 |
| Meditron (Chen et al., 2023) | 0.801 | 0.768 | 0.723 | 0.681 | 0.673 | 0.634 |
| BioMistral (Labrak et al., 2024) | 0.776 | 0.731 | 0.685 | 0.638 | 0.628 | 0.582 |
| Llama-3-Meditron (Sallinen et al., 2025) | 0.795 | 0.756 | 0.712 | 0.668 | 0.658 | 0.619 |
| DoctorGLM (Xiong et al., 2023) | 0.768 | 0.724 | 0.673 | 0.625 | 0.614 | 0.571 |
| GatorTron (Yang et al., 2022) | 0.753 | 0.708 | 0.657 | 0.609 | 0.597 | 0.553 |
| *Multi-Agent Systems* | | | | | | |
| MedAgents (Tang et al., 2024) | 0.823 | 0.789 | 0.751 | 0.708 | 0.692 | 0.653 |
| AgentClinic (Schmidgall et al., 2024) | 0.817 | 0.781 | 0.738 | 0.695 | 0.681 | 0.641 |
| AutoGen (Wu et al., 2023) | 0.804 | 0.765 | 0.724 | 0.679 | 0.665 | 0.624 |
| MetaGPT (Hong et al., 2024) | 0.811 | 0.773 | 0.731 | 0.687 | 0.673 | 0.632 |
| **HetMedAgent (Ours, w/o Clinician)** | **0.866** | **0.844** | **0.801** | **0.757** | **0.727** | **0.719** |

All baselines are provided with the same available multimodal inputs as HetMedAgent.

*Table 2.* Specialist model architecture comparison: Traditional CNNs vs. Transformers on diagnostic quality (BERTScore) and conflict scores ($\delta$, Eq. 5). Best results in **bold**.

| Architecture | Modality | Key Components | BERTScore ↑ | $\delta$ (mean±std) ↓ |
|---|---|---|---|---|
| *Traditional Specialist Models* | | | | |
| ResNet-50-based | ECHO | Seq2Image+Conv+Residual | 0.707 | 0.323±0.077 |
| | ECG | Conv+Residual | 0.658 | |
| EfficientNet-B0-based | ECHO | Seq2Image+MBConv+SE | 0.748 | 0.305±0.091 |
| | ECG | MBConv+SE | 0.691 | |
| *HetMedAgent Specialist Models* | | | | |
| Transformer-based | ECHO | Self-Attn+LSTM+Cross-Attn | **0.800** | **0.279±0.109** |
| | ECG | Conv+Self-Attn+Cross-Attn | **0.717** | |

Seq2Image: Sequence-to-image conversion via adaptive pooling and tensor reshaping; Conv: Convolutional layers; Residual: Residual connections; MBConv: Mobile Inverted Bottleneck Convolution; SE: Squeeze-and-Excitation; Self-Attn: Self-attention; Cross-Attn: Cross-attention; LSTM: Long Short-Term Memory.

tonomous" mode, outputs are recommendations under clinician oversight, not executed orders.

### 3.2.7 CLINICIAN AGENT

The clinician agent $A^P$ plays a dual role in HetMedAgent: (1) providing clinician oversight for high-uncertainty cases where $U(D_{\text{prelim}}) > \theta_P$; (2) serving as a learning mechanism that calibrates intervention threshold $\theta_P$ through operational feedback.

**Adaptive Threshold Calibration.** After each case escalated to $A^P$ is resolved, the system updates the intervention threshold based on clinician feedback:

$$\theta_P \leftarrow \theta_P + 0.001 \cdot (1 - 2m) \tag{12}$$

where $m = 1$ if the clinician modified the decision, 0 otherwise. Accepted decisions increase $\theta_P$ (more autonomy); modifications decrease it (earlier review), balancing efficiency and safety.

### 3.3 Dynamic Collaboration Protocol

The complete protocol (Algorithm 1, Appendix D) proceeds as: (1) orchestrator $\mathcal{O}$ activates specialists based on available modalities; (2) specialists execute in parallel, pro-

ducing findings with confidence scores; (3) the system computes conflict scores (Eq. 5), weights (Eq. 7), and assembles evidence (Eq. 6); (4) reasoning agent $A^r$ generates $D_{\text{prelim}}$ with reasoning chain $\mathcal{R}$; (5) composite uncertainty (Eq. 10) determines whether to output the recommendation or escalate to clinician $A^P$; (6) $\theta_P$ is adaptively calibrated via feedback (Eq. 12). Figure 3 illustrates the complete workflow, demonstrating the end-to-end multi-agent collaborative decision-making process.

## 4 Experiments

### 4.1 Experimental Setup

#### 4.1.1 DATASETS AND TASKS

We evaluate HetMedAgent on a retrospective multimodal cardiovascular clinical decision-making dataset collected from real clinical scenarios, approved by the Institutional Review Board. The dataset comprises 613 test cases from 514 patients, each containing echocardiography (ECHO) reports and ECG images. We assess two complementary aspects: (1) **Specialist Model Diagnosis Generation**: evaluating $A^w_{\text{ECHO}}$ (text-to-text) and $A^w_{\text{ECG}}$ (image-to-text) diagnostic text quality using BERTScore; (2) **HetMedAgent**

*Table 3.* Performance comparison of HetMedAgent with different generalist LLMs across three clinical decision-making tasks. Best in **bold**.

| Generalist LLM | Risk Stratification | | Etiology Prediction | | Severity Assessment | | Average | |
|---|---|---|---|---|---|---|---|---|
| | AUROC ↑ | F1 ↑ | AUROC ↑ | F1 ↑ | AUROC ↑ | F1 ↑ | AUROC ↑ | F1 ↑ |
| GPT-4o | **0.866** | **0.844** | **0.801** | **0.757** | **0.727** | **0.719** | **0.798** | **0.773** |
| Claude-3.5-Sonnet | 0.859 | 0.836 | 0.794 | 0.749 | 0.721 | 0.712 | 0.791 | 0.766 |
| Gemini-2.0-Flash | 0.851 | 0.827 | 0.786 | 0.741 | 0.713 | 0.703 | 0.783 | 0.757 |
| Llama-3.3-70B | 0.843 | 0.819 | 0.778 | 0.732 | 0.706 | 0.695 | 0.776 | 0.749 |
| Qwen-2.5-72B | 0.839 | 0.814 | 0.773 | 0.727 | 0.701 | 0.689 | 0.771 | 0.743 |
| GLM-4 | 0.832 | 0.807 | 0.767 | 0.721 | 0.694 | 0.683 | 0.764 | 0.737 |

All experiments conducted with HetMedAgent framework (w/o clinician intervention). Specialist models ($A_{ECHO}^w$ and $A_{ECG}^w$) remain identical across all configurations; only the generalist LLM component varies.

*Table 4.* Modality ablation within HetMedAgent (w/o clinician). Best in **bold**.

| Configuration | Average across 3 tasks | | |
|---|---|---|---|
| | AUROC ↑ | F1 ↑ | *p*-value |
| GPT-4o | 0.671 | 0.625 | - |
| + $A_{ECHO}^w$ | 0.752 | 0.711 | <0.001 |
| + $A_{ECG}^w$ | 0.734 | 0.692 | <0.001 |
| + Both specialists | **0.798** | **0.773** | <0.001 |

Metrics averaged across risk stratification, etiology prediction, and severity assessment. *p*-values computed using Bootstrap resampling test (10,000 iterations) comparing each configuration against GPT-4o baseline.

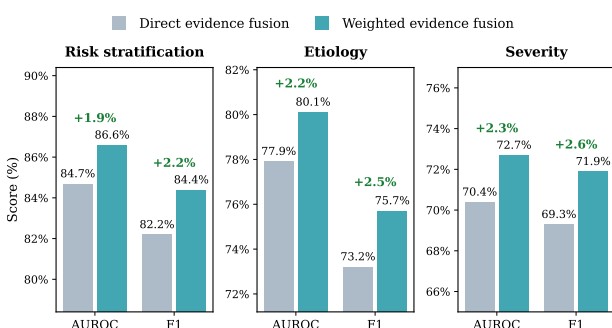

*Figure 4.* Performance comparison between weighted and direct evidence fusion strategies across clinical decision-making tasks.

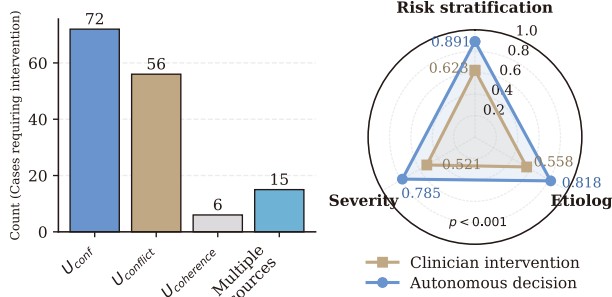

*Figure 5.* (**Left**) Uncertainty decomposition analysis (fixed threshold $\theta_P = 0.5$). (**Right**) Comparing F1 scores between autonomous decisions and cases requiring clinician intervention across clinical decision-making tasks (fixed threshold $\theta_P = 0.5$; Mann–Whitney $U$ test, $p < 0.001$).

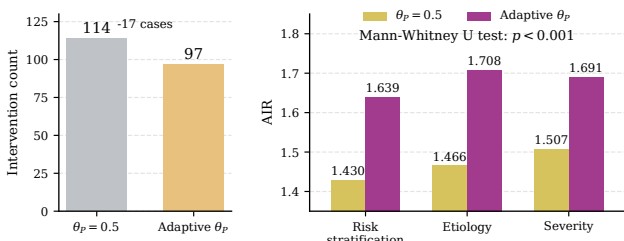

*Figure 6.* Impact of adaptive threshold calibration ($\theta_P = 0.5$). (**Left**) Number of cases requiring clinician intervention under different thresholds (bar chart). (**Right**) Autonomous-to-Intervention F1 Ratio (AIR) comparison across clinical tasks (bar chart).

**Clinical Decision-Making Tasks**: evaluating admission risk stratification, etiology prediction, and severity assessment using AUROC and F1-Score. Both Transformer-based specialist models output diagnostic text with confidence scores. In our implementation, we use GPT-4o as the generalist LLM serving the orchestration and reasoning roles. Dataset details, specialist model architectures, and training procedures are provided in Appendices E and F.

### 4.1.2 BASELINE METHODS

We compare HetMedAgent with two categories of baselines: **(1) single model methods**, including medical LLMs (PMC-LLaMA (Wu et al., 2024), Meditron (Chen et al., 2023), BioMistral (Labrak et al., 2024), Llama-3-Meditron (Sallinen et al., 2025), DoctorGLM (Xiong et al., 2023), GatorTron (Yang et al., 2022)) and specialist models (e.g., ResNet (He et al., 2016), EfficientNet (Tan &

Le, 2019) for ECHO and ECG diagnosis); **(2) multi-agent systems**, including MedAgents (Tang et al., 2024), Agent-Clinic (Schmidgall et al., 2024), AutoGen (Wu et al., 2023), and MetaGPT (Hong et al., 2024). More details on baseline configurations are provided in Appendix G.

### 4.2 Main Results

Table 1 shows HetMedAgent consistently outperforms both single-model and multi-agent baselines across all three tasks. Table 2 demonstrates our Transformer-based specialist models achieve superior diagnostic quality and lower conflict scores than CNN-based models.

**Key Observations.**

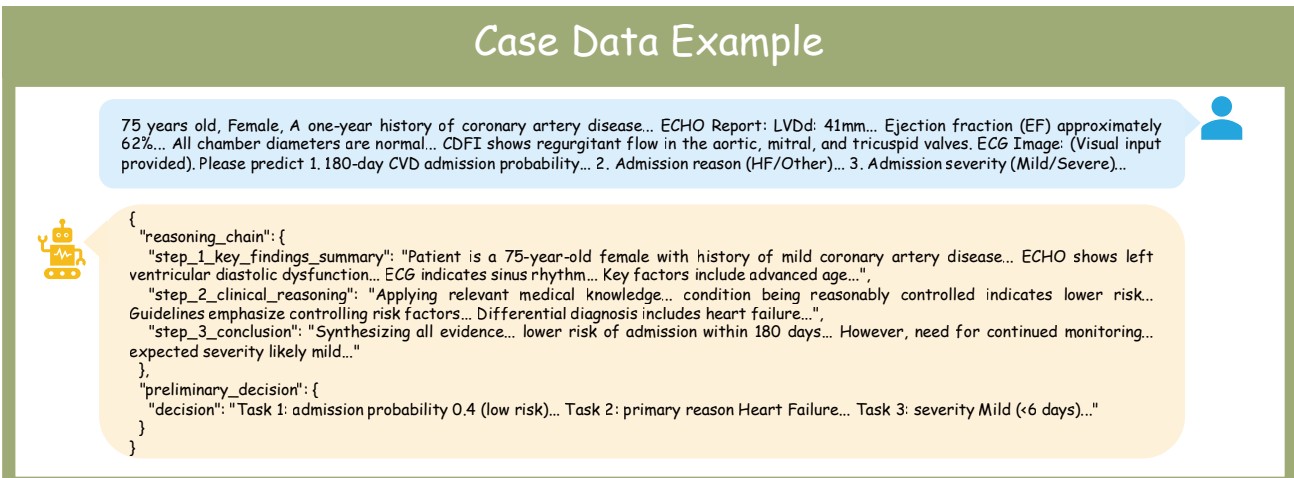

*Figure 7.* Case study of HetMedAgent's autonomous decision-making from multimodal inputs to final clinical decisions.

HetMedAgent achieves average improvements of 6.6% in AUROC and 7.9% in F1 score over the best single-model baseline (Meditron), and 4.3% in AUROC and 5.7% in F1 score over the best multi-agent system (MedAgents). Our Transformer-based specialist models ($A_{\text{ECHO}}^w$ and $A_{\text{ECG}}^w$) substantially outperform traditional CNN-based approaches (ResNet-50-based, EfficientNet-B0-based), with BERTScore gains of 9.3%/5.9% over ResNet-50 and 5.2%/2.6% over EfficientNet-B0 for ECHO/ECG, while reducing mean conflict scores by 4.4% and 2.6%, respectively.

### 4.3 Ablation Studies

We conduct systematic ablation studies to analyze the contribution of each component in HetMedAgent.

**Generalist LLM Comparison.** Table 3 evaluates HetMedAgent with different generalist LLMs (specialist models fixed). GPT-4o achieves optimal performance (0.798 AUROC, 0.773 F1), followed by Claude-3.5-Sonnet (0.791, 0.766) and Gemini-2.0-Flash (0.783, 0.757). Llama-3.3-70B and Qwen-2.5-72B show competitive results with 2.2% and 2.7% AUROC gaps, validating compatibility with diverse LLM backends.

**Modality Analysis.** Table 4 shows the impact of different modality combinations, reporting average performance across the three tasks. The GPT-4o baseline achieves 0.671 AUROC and 0.625 F1. Adding $A_{\text{ECHO}}^w$ improves performance to 0.752 AUROC (+8.1%) and 0.711 F1 (+8.6%), while adding $A_{\text{ECG}}^w$ yields 0.734 AUROC (+6.3%) and 0.692 F1 (+6.7%). The full system with both specialist models achieves the best results: 0.798 AUROC (+12.7%) and 0.773 F1 (+14.8%), demonstrating that ECHO and ECG specialists provide complementary information. Per-task performance differences across modality combinations are further visualized in Figure 12 in Appendix H.

**Weighted Evidence Fusion.** Figure 4 compares direct ev-

*Table 5.* Computational efficiency comparison on NVIDIA RTX 4090 GPU with batch size of 1. Inference time, FLOPs, and parameters for specialist models and complete system.

| Component | Time/s ↓ | FLOPs/G ↓ | Params/M ↓ |
|---|---|---|---|
| $A_{\text{ECHO}}^w$ | 1.130 | 2.411 | 9.703 |
| $A_{\text{ECG}}^w$ | 2.111 | 2.487 | 46.307 |
| HetMedAgent (w/o Clinician) | 26.684 | 4.898 | 56.010 |

Time: Inference time in seconds; FLOPs: Floating point operations in billions; Params: Model parameters in millions. FLOPs and Params count local specialist models only; generalist LLM computation is performed via API and excluded.

idence fusion (equal weighting of all specialist outputs) against our weighted evidence fusion strategy (Eq. 6) that dynamically adjusts weights based on specialist confidence and conflict scores. Weighted fusion consistently outperforms direct fusion across all three tasks, achieving improvements of 2.1% in AUROC and 2.4% in F1-Score on average.

**Uncertainty Quantification.** Among 613 test cases, 114 (18.6%) exceeded the fixed threshold $\theta_P = 0.5$, triggering clinician intervention. Figure 5 (left) shows the uncertainty source distribution; the right panel compares F1 between autonomous and escalated cases. Autonomous cases exhibit consistently higher F1 (Mann-Whitney U, $p < 0.001$), validating the effectiveness of uncertainty-based routing in screening difficult cases for escalation.

**Adaptive Threshold Calibration.** We first define the Autonomous-to-Intervention F1 Ratio (AIR) as a quantitative metric of system decision quality, computed as:

$$\text{AIR} = \frac{F1_{\text{autonomous}}}{F1_{\text{intervention}}} \tag{13}$$

Higher ratios indicate the system's ability to accurately identify high-uncertainty cases, routing truly difficult cases to clinician intervention.

To validate adaptive threshold calibration, we initialized the intervention threshold at $\theta_P = 0.5$ and randomly shuffled

613 test cases to simulate sequential clinical decisions. Using ground truth as intervention feedback, the system dynamically adjusts $\theta_P$ per Eq. 12, triggering intervention in 97 cases (15.8%)—17 fewer cases than the fixed threshold, whose intervention rate was 18.6%. Figure 6 shows adaptive calibration improves average AIR from 1.468 to 1.679 (Mann-Whitney U test, $p < 0.001$), demonstrating precise boundary case identification through clinician feedback.

### 4.4 Efficiency and Cost Analysis

Table 5 presents efficiency metrics on NVIDIA RTX 4090 GPU. The $A_{\text{ECHO}}^w$ and $A_{\text{ECG}}^w$ specialists achieve 1.130s and 2.111s inference with $\sim$2.5G FLOPs and compact parameters (9.7M and 46.3M). HetMedAgent processes cases in 26.684s, suitable for non-emergency scenarios. At GPT-4o API pricing ($0.0025 per 1K input, $0.01 per 1K output tokens), each case uses $\sim$2000 input and 500 output tokens, costing $0.01 per case.

### 4.5 Case Study

We present a representative case demonstrating HetMedAgent's decision-making process.

**Successful Autonomous Decision.** Figure 7 illustrates the actual decision-making process with inputs (patient clinical information, ECHO report, ECG image, and task definition) and outputs (the step-by-step reasoning chain, the final clinical decision outputs). The complete routing trace, including orchestrator dispatch, specialist outputs, conflict-aware evidence fusion, uncertainty assessment, and the final autonomous recommendation, is provided in Appendix I.

### 4.6 Subgroup Fairness Analysis

Disaggregated analysis across gender and age subgroups (Appendix J) shows largely stable performance: no significant gender or age differences in most tasks (Fisher's exact test, $p > 0.05$), with only etiology AUC showing a marginal gap ($p=0.026$).

### 4.7 Cross-Domain Validation

To evaluate generalizability, we applied HetMedAgent to the IU X-Ray dataset (chest radiology, 668 test cases) with a chest X-ray specialist ($A_{\text{CXR}}^w$). HetMedAgent achieved 0.820 AUROC and 0.537 F1 on an acute/non-acute task, outperforming a domain-specific multimodal ViT-BERT fusion baseline (0.783 AUROC, 0.468 F1), confirming effective transfer across clinical domains (Appendix K). Appendix L further empirically validates that the reasoning agent is meaningfully sensitive to weight annotations rather than merely ignoring them during synthesis. Additional uncertainty-component ablations are reported in Appendix M.

## 5 Conclusions and Future Directions

We presented HetMedAgent, a heterogeneous medical multi-agent framework that orchestrates generalist LLMs, domain-specific specialist models, and human clinicians. Experiments on three cardiovascular tasks demonstrate that generalist-specialist synergy significantly outperforms both monolithic medical LLMs and existing multi-agent systems, validating the irreplaceable value of domain-specific models. The conflict-aware evidence fusion and uncertainty-based routing enable reliable autonomous operation while escalating genuinely difficult cases for clinician review.

**Limitations.** Key limitations include: (1) the generation confidence $c_i$ captures token-level prediction certainty rather than clinical correctness; (2) routing relies solely on epistemic uncertainty without clinical severity weighting; (3) adaptive calibration uses simulated (ground-truth) rather than real clinician feedback; (4) evaluation is primarily single-institution (613 cardiovascular cases), though cross-domain results (Appendix K) are encouraging; (5) the Age$\geq$85 subgroup is small ($n$=47) and comorbidity-level analysis was not conducted; (6) commercial LLM APIs raise privacy concerns despite de-identification; (7) the text-only inter-agent interface may lose nuanced structural or spatial features that continuous representations could preserve.

**Future Directions.** Promising extensions include: (1) multi-faceted confidence decomposition capturing measurement reliability and diagnostic specificity; (2) risk-aware routing that modulates $\theta_P$ with task-specific severity scores ($\theta_P^{\text{effective}} = \theta_P \cdot g(\text{severity})$) and systematic exploration of learned or task-adaptive uncertainty weights $\lambda$ beyond the equal-contribution setting; (3) momentum-based adaptive calibration ($\theta_P \leftarrow \theta_P + \eta(\beta \cdot \Delta_{\text{prev}} + (1 - \beta) \cdot \Delta_{\text{current}})$) with multi-clinician consensus, exclusion of highly divergent responses, and bounded update magnitudes to handle real feedback noise; (4) per-modality confidence calibration via Platt scaling; (5) local deployment with open-weights models for privacy-preserving operation; (6) cross-domain extension to oncology, neurology, and emergency medicine with additional specialist modalities, including systematic validation of cases where the number of invoked specialists satisfies $k > 2$ to assess cross-specialist conflict modelling and evidence fusion under richer multi-modal collaboration; (7) a hybrid inter-agent interface where specialists produce both structured textual summaries (for interpretability) and continuous embeddings (for multimodal fusion), enabling the reasoning module to jointly consume both representations; (8) clinician interaction optimization through interactive visualization dashboards and formal usability studies (e.g., NASA-TLX) to quantify cognitive load under real deployment conditions.

## Acknowledgments

This work was supported by the National Natural Science Foundation of China (No. 62371138) and the National Key R&D Program of China (No. 2024YFC2418500).

## Impact Statement

**Ethics and Accountability:** HetMedAgent operates as a Clinical Decision Support System (CDSS) under clinician oversight. Human clinicians retain exclusive authority over final diagnoses, treatment orders, patient communication, and professional accountability. Even in "autonomous" mode, system outputs are recommendations; the clinician agent $A^P$ has unconditional override authority. This study was approved by the Medical Ethics Committee of Xinghua City People's Hospital affiliated to Yangzhou University (Approval No. JSXHRYLL-YL-202430). Informed consent was waived for this retrospective study of de-identified data.

**Privacy:** The modular architecture supports replacement of commercial APIs with locally deployed open-weights models for physical data isolation under HIPAA/GDPR. All patient data in this study was fully de-identified prior to processing.

**Societal Impact:** HetMedAgent could improve decision-making accessibility in resource-limited settings. We advocate deployment frameworks that preserve clinician agency rather than reducing practitioners to passive validators of AI outputs.

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

# A   Notations

Table 6. Notations and Definitions

| Notation | Definition |
|---|---|
| $C = \{V, I\}$ | Patient case comprising clinical information $V$ and multimodal examination data $I$. |
| $V$ | Clinical information including age, gender, chronic disease history, treatment history, and symptoms. |
| $I$ | Multimodal examination data comprising imaging, physiological signals, etc. |
| $I_i$ | Individual modality data within $I$ (e.g., ECG, echocardiography). |
| $D = \{d_1, \ldots, d_k\}$ | Set of clinical decisions generated by the system. |
| $Q(D, C)$ | Decision quality metric evaluating the correctness and utility of decisions $D$ for case $C$. |
| $M(C)$ | Single-model approach generating decisions directly from case $C$. |
| $\mathcal{O}$ | Orchestrator agent built on generalist LLMs for task coordination and agent management. |
| $A_i^w$ | Specialist agent for analyzing modality-specific data (e.g., $A_{\text{ECHO}}^w$, $A_{\text{ECG}}^w$). |
| $A^r$ | Reasoning agent built on generalist LLMs for generating preliminary decisions. |
| $A^P$ | Clinician agent providing human oversight and final decision authority. |
| $F_i^w$ | Findings output from specialist agent $A_i^w$ including diagnosis and confidence. |
| $c_i \in [0, 1]$ | Generation confidence associated with specialist agent $A_i^w$'s findings. |
| $\text{Context}(C)$ | Comprehensive context including $V(C)$, $H(C)$, $L(C)$, and $D(C)$. |
| $V(C)$ | Patient clinical information stored in memory module. |
| $H(C)$ | Interaction history stored in memory module. |
| $L(C)$ | Modality registry for dynamic agent activation. |
| $D(C)$ | Clinical tasks identified from case $C$. |
| $\psi_{\text{task}}$ | Structured prompt for task identification by orchestrator $\mathcal{O}$. |
| $\psi_{\text{reason}}$ | Structured prompt for clinical reasoning by reasoning agent $A^r$. |
| $\psi_i^w$ | Structured prompt for specialist agent $A_i^w$. |
| $\varphi : L \times D \rightarrow 2^{A^w}$ | Mapping function from modalities and tasks to activated specialist agent subsets. |
| $A_{\text{active}}$ | Set of activated specialist agents for a given case. |
| $\delta_i$ | Conflict score measuring disagreement of specialist $i$ with other specialists (Eq. 5). |
| $\text{sim}(\cdot, \cdot)$ | Cosine similarity function between embeddings. |
| $w_i$ | Weight assigned to specialist agent $A_i^w$ for evidence fusion (Eq. 7). |
| $D_{\text{prelim}}$ | Preliminary decision generated by reasoning agent $A^r$. |
| $\mathcal{R} = \{s_1, \ldots, s_m\}$ | Reasoning chain with $m$ steps generated by $A^r$. |
| $s_t$ | Individual reasoning step at position $t$ in reasoning chain. |
| $U(D_{\text{prelim}})$ | Comprehensive uncertainty score for preliminary decision (Eq. 10). |
| $U_{\text{conf}}$ | Confidence-based uncertainty component: $U_{\text{conf}} = 1 - \max_i(c_i)$. |
| $U_{\text{conflict}}$ | Conflict-based uncertainty component: $U_{\text{conflict}} = \frac{1}{k} \sum_i \delta_i$. |
| $U_{\text{coherence}}$ | Reasoning coherence-based uncertainty component (Eq. 9). |
| $\lambda_{\text{conf}}, \lambda_{\text{conflict}}, \lambda_{\text{coherence}}$ | Non-negative contribution weights for uncertainty components in Eq. 10, normalised to sum to 1. |
| $\theta_P$ | Threshold for triggering clinician intervention based on uncertainty. |
| $D_{\text{output}}$ | Final clinical decision output by the system. |
| AIR | Autonomous-to-Intervention F1 Ratio measuring system decision quality (Eq. 13). |
| $F1_{\text{autonomous}}$ | F1 score for cases handled autonomously by the system. |
| $F1_{\text{intervention}}$ | F1 score for cases requiring clinician intervention. |
| $k$ | Total number of specialist agents activated for a given case. |
| $m$ | Number of reasoning steps in reasoning chain $\mathcal{R}$. |
| $\text{LLM}_{\mathcal{O}}$ | Large language model serving as orchestrator agent. |
| $\text{LLM}_{A^r}$ | Large language model serving as reasoning agent. |

# B   Additional Visualizations

This appendix provides additional prompt visualizations (Figure 8 and Figure 9) supporting the analysis in the main paper.

# C   Conflict Detection: Implementation and Controlled Examples

**Conflict Score Computation.** The conflict score quantifies semantic inconsistency between specialist outputs rather than superficial lexical overlap. Given two specialist outputs $F_i$ and $F_j$, we encode them into normalized embeddings $\mathbf{e}_i$ and $\mathbf{e}_j$ using a publicly available PubMedBERT-based biomedical bi-encoder, and compute their cosine similarity:

$$s_{ij} = \cos(\mathbf{e}_i, \mathbf{e}_j).$$

The pairwise conflict score is then defined as

$$\delta_{ij} = 1 - \frac{1 + s_{ij}}{2},$$

and $U_{\text{conflict}} = \frac{1}{k} \sum_i \delta_i$ is used as the cross-specialist conflict component of the composite uncertainty score (Eq. 10).

**Shared-Direction Adjustment.** To mitigate consistent phrasing biases across modalities, we apply: $\tilde{\mathbf{e}}_i = \text{norm}(\mathbf{e}_i + \alpha\boldsymbol{\mu})$, where $\boldsymbol{\mu}$ is estimated as the mean embedding direction across all specialist outputs in a calibration set, and $\alpha$ is tuned on the validation split.

Figure 10 presents two controlled extreme examples designed to probe the boundary cases of the detection mechanism. The results demonstrate that the proposed approach reliably distinguishes terminology variation from genuine semantic conflict, validating its use as the $U_{\text{conflict}}$ component of the composite uncertainty score.

## D  Decision-Making Protocol

This appendix provides the complete pseudo-code for HetMedAgent's end-to-end decision-making protocol (Algorithm 1), corresponding to the procedure described in the main paper.

---

**Algorithm 1** HetMedAgent Decision-Making Protocol

---

**Require:** Patient case $C$ with multimodal data
**Ensure:** Clinical decision $D_{\text{output}}$
 1: $\{A_1^w, \ldots, A_k^w\} \leftarrow \mathcal{O}(\text{Context}(C))$          $\triangleright$ Activate specialists
 2: **for** each specialist $A_i^w$ in parallel **do**
 3:      $F_i^w, c_i \leftarrow A_i^w(I_i, \psi_i^w)$          $\triangleright$ Analyze modality $I_i$
 4: **end for**
 5: Compute conflict scores $\{\delta_i\}$ via Eq. 5
 6: Compute weights $\{w_i\}$ via Eq. 7 using $\{c_i, \delta_i\}$
 7: Assemble $\text{Input}_{\text{reason}}$ via Eq. 6
 8: $D_{\text{prelim}}, \mathcal{R} \leftarrow \text{LLM}_{A^r}(\psi_{\text{reason}}, \text{Input}_{\text{reason}})$          $\triangleright$ Eq. 8
 9: Compute uncertainty $U(D_{\text{prelim}})$ via Eq. 10
10: **if** $U(D_{\text{prelim}}) \leq \theta_P$ **then**
11:      $D_{\text{output}} \leftarrow D_{\text{prelim}}$          $\triangleright$ Autonomous recommendation
12: **else**
13:      $D_{\text{output}} \leftarrow A^P(D_{\text{prelim}}, \mathcal{R})$          $\triangleright$ Clinician review
14:      Update $\theta_P$ via Eq. 12          $\triangleright$ Adaptive calibration
15: **end if**
16: **return** $D_{\text{output}}$

---

## E  Dataset Details

We constructed three datasets for specialist model training and comprehensive evaluation of HetMedAgent. Table 7 summarizes the statistics of all three datasets.

*Table 7.* Dataset statistics for specialist model training and evaluation.

| Dataset | Patients | Cases | Modality |
|---|---|---|---|
| $A_{\text{ECHO}}^w$ Training | 5,783 | 10,400 | ECHO Reports |
| $A_{\text{ECG}}^w$ Training | 4,326 | 10,000 | ECG Images |
| Multimodal Test | 514 | 613 | ECHO + ECG |

**Data Collection and Preprocessing.** The $A_{\text{ECHO}}^w$ training set contains structured echocardiography reports paired with expert-annotated diagnostic text. Each report includes standardized measurements of cardiovascular structure and function. The $A_{\text{ECG}}^w$ training set comprises 12-lead ECG images paired with expert-annotated diagnostic interpretations. All ECG images were standardized to 224×224 resolution with consistent formatting.

**Data Splitting Strategy.** Both $A_{\text{ECHO}}^w$ and $A_{\text{ECG}}^w$ training datasets were randomly split into training (80%) and validation (20%) subsets at the patient level to ensure no patient appears in both subsets. The multimodal test set was independently collected from real clinical scenarios and contains patients completely distinct from those in the training datasets, ensuring rigorous evaluation without data leakage. This multimodal test set serves dual purposes: (1) evaluating specialist model diagnostic generation quality using BERTScore, and (2) assessing HetMedAgent's clinical decision-making performance on admission risk stratification, etiology prediction, and severity assessment tasks.

**Demographic Distribution.** Figure 11 illustrates the age and gender distribution of the multimodal test set. The age distribution spans from 21 to 93 years with a mean of 70.09±11.56 years. Gender distribution is balanced with 55.1% male and 44.9% female patients, reflecting representative clinical populations.

**Clinical Task Definitions.** The multimodal test set supports evaluation of three clinical decision-making tasks: (1) *Admission Risk Stratification*: Binary classification predicting the probability of cardiovascular disease-related hospital admission within 180 days post-baseline. The model outputs a risk score in [0,1], with 0.5 as the decision threshold (low-risk: $< 0.5$, high-risk: $\geq 0.5$). This task assesses the system's ability to identify patients requiring close monitoring and preventive interventions based on baseline clinical data. (2) *Etiology Prediction*: Binary classification identifying the primary etiology of cardiovascular disease-related admissions within 180 days post-baseline. The model classifies admissions into two categories: heart failure versus other cardiovascular causes. This task evaluates the system's capability to predict the underlying pathophysiological mechanism driving hospitalization. (3) *Severity Assessment*: Binary classification predicting the severity of future cardiovascular disease-related admissions (not limited to 180 days). Severity is defined by length of stay: mild cases ($< 6$ days) versus severe cases ($\geq 6$ days). This task measures the system's ability to forecast resource utilization and clinical complexity. Ground truth labels for all tasks were established through retrospective chart review and consensus evaluation by board-certified cardiologists.

Table 8 summarizes the label distribution across all three clinical decision-making tasks in the multimodal test set.

*Table 8.* Label distribution of clinical decision-making tasks in the multimodal test set ($n$=613).

| Task | Category | Cases (Percentage) |
|---|---|---|
| Admission Risk Stratification | High-risk ($\leq$180 days) | 305 (49.76%) |
| | Low-risk ($>$180 days) | 308 (50.24%) |
| Etiology Prediction | Heart failure | 113 (37.05%) |
| | Other cardiovascular causes | 192 (62.95%) |
| Severity Assessment | Mild ($<$6 days) | 300 (48.94%) |
| | Severe ($\geq$6 days) | 313 (51.06%) |

**Note:** Etiology Prediction covers only the $n$=305 high-risk cases with CVD-related admissions within 180 days post-baseline (corresponding to the High-risk group in Admission Risk Stratification); the remaining 308 low-risk cases do not have an etiology label. Risk Stratification and Severity Assessment cover the full test set ($n$=613).

# F   Specialist Model Architectures and Training

We developed two Transformer-based specialist models for echocardiography report interpretation and ECG image analysis. Table 9 summarizes the architectural and training hyperparameters of both models.

$A_{\mathbf{ECHO}}^w$ **Model Architecture.** $A_{\mathrm{ECHO}}^w$ adopts an Encoder-Session-Decoder architecture for text-to-text generation. The encoder and decoder are Transformer-based, while the session layer employs an LSTMCell with attention mechanism to maintain temporal context across sequences. The architecture supports weight sharing between target word embeddings and output projection layer.

$A_{\mathbf{ECG}}^w$ **Model Architecture.** $A_{\mathrm{ECG}}^w$ combines a CNN-based image encoder with Transformer encoder-decoder for image-to-text generation. The CNN encoder extracts spatial features from ECG images ($224 \times 224$) through 4 convolutional blocks, producing a $7 \times 7$ feature map (49 visual tokens). These features are then processed by the Transformer encoder and decoded by the Transformer decoder through cross-attention mechanisms.

**Training Procedure.** Both models were trained using Adam optimizer with warmup learning rate schedule for the first 4,000 steps. Label smoothing with factor 0.1 was applied to both models. Training continued with early stopping based on validation accuracy. Data augmentation for $A_{\mathrm{ECG}}^w$ included random cropping, brightness adjustment ($\pm0.2$), and contrast adjustment ($\pm0.2$) to improve robustness.

**Loss Function.** The training objective is cross-entropy loss with label smoothing for text generation. Formally, the loss is $\mathcal{L}_{\mathrm{CE}}$ with smoothing factor $\epsilon = 0.1$, which helps prevent overconfidence and improves model generalization. The loss ignores padding tokens during backpropagation.

**Generation Confidence Estimation.** During inference, generation confidence scores $c_i \in [0, 1]$ are computed from per-token softmax probabilities. Specifically, for a generated sequence with tokens $\{y_1, \ldots, y_L\}$, the generation confidence is calculated as the geometric mean of per-token softmax probabilities (equivalently, inverse perplexity): $c_i = (\prod_{t=1}^{L} p(y_t))^{1/L}$, where $p(y_t)$ is the softmax probability of token $y_t$. These generation confidence scores are used by

*Table 9.* Architectural hyperparameters of specialist models.

| Hyperparameter | $A_{\text{ECHO}}^{w}$ | $A_{\text{ECG}}^{w}$ |
|---|---|---|
| *Architecture* | | |
| Number of layers | 3 | 6 |
| Model dimension ($d_{\text{model}}$) | 256 | 512 |
| Word embedding dim ($d_{\text{word}}$) | 256 | 512 |
| FFN inner dimension ($d_{\text{inner}}$) | 1024 | 2048 |
| Number of attention heads | 8 | 8 |
| Key/Value dimension ($d_k/d_v$) | 64 | 64 |
| LSTM hidden dimension | 512 | — |
| Max sequence length | 300 | 50 |
| Input modality | Text | Image ($224 \times 224$) |
| CNN encoder blocks | — | 4 |
| Visual tokens | — | 49 ($7 \times 7$) |
| *Training* | | |
| Optimizer | Adam | Adam |
| Learning rate | 1e-3 | 1e-4 |
| Adam $\beta_1$, $\beta_2$ | 0.9, 0.98 | 0.9, 0.98 |
| Warmup steps | 4,000 | 4,000 |
| Batch size | 16 | 8 |
| Dropout rate | 0.1 | 0.1 |
| Label smoothing | 0.1 | 0.1 |
| Max epochs | 50 | 100 |
| Early stopping patience | 5 | 10 |
| Weight sharing | Embedding-Proj | Embedding-Proj |

Embedding-Proj = weight sharing between target word embeddings and output projection layer.

HetMedAgent's uncertainty-based routing mechanism to determine when to trigger clinician intervention.

# G   Baseline Configurations

We provide detailed configurations for all baseline methods evaluated in our experiments.

**Single Model Baselines.** We evaluate six single model baselines: (1) *PMC-LLaMA* (Wu et al., 2024): A medical LLM adapted via biomedical knowledge injection and medical instruction tuning for medical QA and clinical reasoning. (2) *Meditron* (Chen et al., 2023): We use the publicly released Meditron-70B weights from EPFL, a medical LLM pre-trained on large-scale medical literature and clinical data. (3) *BioMistral* (Labrak et al., 2024): A 7B parameter open-source medical LLM fine-tuned from Mistral-7B (Jiang et al., 2023) on PubMed abstracts and medical guidelines. (4) *Llama-3-Meditron* (Sallinen et al., 2025): An open-weight suite of medical LLMs based on Llama-3.1, adapted for clinical reasoning and medical question answering. (5) *DoctorGLM* (Xiong et al., 2023): A Chinese medical LLM fine-tuned for clinical dialogue and medical reasoning tasks. (6) *GatorTron* (Yang et al., 2022): A large language model pre-trained on extensive electronic health records for clinical text understanding. (7) *Specialist Baseline Models for ECHO*: We train ResNet-50-based (He et al., 2016) and EfficientNet-B0-based (Tan & Le, 2019) models on ECHO report classification tasks. These models convert text sequences to pseudo-images via adaptive pooling and reshaping (Seq2Image), then apply convolutional feature extraction. In contrast, our $A_{\text{ECHO}}^{w}$ uses an Encoder-Session-Decoder architecture with Transformer and LSTM components for text generation with confidence estimation. (8) *Specialist Baseline Models for ECG*: We train ResNet-50-based (He et al., 2016) and EfficientNet-B0-based (Tan & Le, 2019) models on ECG image classification tasks. In contrast, our $A_{\text{ECG}}^{w}$ uses a CNN-based image encoder with Transformer encoder-decoder for diagnostic text generation. These specialist baseline models operate on single modalities without access to complementary data.

**Multi-Agent Baselines.** We compare against four multi-agent systems: (1) *MedAgents* (Tang et al., 2024): A collaborative multi-agent framework where LLMs work together for zero-shot medical reasoning, with agents assigned different roles such as medical knowledge retrieval, reasoning coordination, and answer synthesis. (2) *AgentClinic*: Following Schmidgall et al. (2024), we implement a multi-agent system where multiple GPT-4 (Achiam et al., 2023) instances role-play as different medical specialists (cardiologist, imaging specialist, general practitioner). Agents engage in dialogue to reach consensus on clinical decisions. We use the same prompting strategies and hyperparameters as reported in the original paper. (3) *AutoGen*: We adapt the AutoGen framework (Wu et al., 2023) for medical decision-making, where multiple LLM agents with different roles (data analyst, domain expert, critic) collaborate through structured conversations to reach

clinical decisions. (4) *MetaGPT*: We adapt the MetaGPT framework (Hong et al., 2024) by assigning LLM agents specific roles in the medical decision pipeline (requirement analysis, diagnostic reasoning, decision integration) with standardized operating procedures for agent collaboration.

## H  Supplementary Results

Figure 12 presents the task-level F1 scores and AUC metrics for different modal configurations. The results demonstrate that our multimodal approach consistently outperforms single-modal baselines across all three clinical tasks (admission risk stratification, etiology prediction, and severity assessment), confirming the complementary nature of echocardiography and ECG modalities in clinical decision-making.

## I  Complete Routing Case

Figure 13 provides the full routing process for the representative autonomous case shown in Figure 7. It details how the orchestrator assigns available modalities to specialist agents, how modality-specific findings are transformed into weighted evidence, how uncertainty is computed for routing, and how the final clinical decision is produced without clinician escalation.

## J  Extended Subgroup Fairness Analysis

Figure 14 presents the full disaggregated performance results on the cardiovascular test set ($n = 613$). The analysis covers three clinical decision-making tasks: 180-day CVD admission risk prediction, heart failure detection, and admission severity classification. Subgroups are defined by sex (Male/Female) and age group following ACC/AHA risk stratification ($<65$, 65–74, 75–84, $\geq85$ years). Statistical significance of between-subgroup differences was assessed using Fisher's exact test.

## K  Cross-Domain Validation: IU X-Ray Experiment

To validate HetMedAgent's generalizability, we conducted experiments on the publicly available IU X-Ray dataset. As summarized in Table 10, this setting differs from the cardiovascular setting across key experimental dimensions, including clinical specialty (chest radiology), imaging type (chest radiographs), and report format (free-text narrative).

**Dataset.** The IU X-Ray dataset contains 3,337 valid cases after filtering, split into 2,335/334/668 for train/validation/test at the patient level (Table 11). We derived a binary acuity label from the COMPARISON field in the radiology XML: cases indicating urgent or interval change are labelled acute (urgent management required), and stable or unchanged findings are labelled non-acute (no urgent management required). The COMPARISON field is used exclusively for label construction and is not provided as input to HetMedAgent, thereby preventing data leakage. The test set contains 67 acute (10.0%) and 601 non-acute (90.0%) cases (Table 12). The full data preprocessing pipeline is described in Table 13.

**Specialist Configuration.** We integrated a chest X-ray specialist ($A_{\text{CXR}}^w$) that shares the CNN+Transformer architecture of $A_{\text{ECG}}^w$ but extends the CNN encoder to accept dual-view (frontal + lateral) CXR images, doubling the visual token count from 49 to 98; full architectural and training hyperparameters are listed in Table 14. The specialist is trained on the IU X-Ray training split to generate radiology findings from paired CXR images. With a single specialist, $U_{\text{conflict}}$ is not applicable; the composite uncertainty estimate therefore reduces to $U_{\text{conf}}$ and $U_{\text{coherence}}$. The complete end-to-end experimental pipeline from specialist training to final evaluation is detailed in Table 15.

**Baseline.** The ViT-BERT baseline (Stage 6, Table 15) uses a Vision Transformer (ViT-B/16, pretrained on ImageNet) to encode dual-view CXR images and BERT-base to encode the INDICATION field; the two representations are concatenated and fed into a two-layer MLP with ReLU activation for binary acuity classification. The model is trained on the same 2,335 training samples with Adam (lr $= 10^{-4}$), batch size 8, early stopping (patience $= 10$), and focal loss ($\gamma = 2$, $\alpha = 0.75$) to address class imbalance.

**Results.** Table 16 shows that HetMedAgent outperforms the ViT-BERT baseline across all metrics, achieving improvements of 3.7% in AUROC (0.820 vs. 0.783) and 6.9% in F1 (0.537 vs. 0.468). Notably, the recall improvement from 0.716 to 0.761 is clinically meaningful: in a dataset with only 10% acute cases, HetMedAgent correctly identifies a larger proportion of patients requiring urgent intervention, directly reducing the risk of missed critical diagnoses. The accuracy gain (0.837 → 0.868) further demonstrates that this improved sensitivity does not come at the cost of increased false positives. These results confirm that the heterogeneous multi-agent architecture, including the orchestrator, reasoning agent, and uncertainty-based routing logic, transfers effectively to a clinical domain that differs substantially from the original cardiovascular setting in specialty, imaging modality, text format, and number of specialists.

*Table 10.* Comprehensive cross-domain comparison between the original cardiovascular and supplementary chest radiology experiments.

| Dimension | Original (Cardiovascular) | Supplementary (Chest Radiology) |
|---|---|---|
| ***Clinical Setting*** | | |
| Clinical specialty | Cardiovascular medicine | General chest radiology |
| Imaging modality | 12-lead ECG waveform images | Chest radiographs (PA/AP + lateral) |
| Text modality | ECHO reports (structured / semi-structured) | Radiology findings (free-text narrative) |
| ***Specialist Configuration*** | | |
| No. of specialist models | $2\ (A_{\text{ECHO}}^w + A_{\text{ECG}}^w)$ | $1\ (A_{\text{CXR}}^w)$ |
| Specialist paradigm | Text-to-text + Image-to-text | Image-to-text (dual-view input) |
| Clinical context input | Age, sex, medical history, treatment history, symptoms | INDICATION from radiology XML |
| ***Task Design*** | | |
| Downstream task(s) | 3 tasks: admission risk, etiology, severity | 1 task: acute vs. non-acute |
| Task type | Binary classification $\times 3$ | Binary classification $\times 1$ |
| ***Dataset*** | | |
| Dataset source | Private retrospective cohort (IRB-approved) | Public IU X-Ray / Open-i |
| Dataset availability | Restricted | Publicly available |
| Specialist training data | ECHO: 10,400 reports (80/20 train/val); ECG: 10,000 images (80/20 train/val) | CXR: 2,669 image–report pairs (70/10 train/val) |
| Test set size | 613 cases / 514 patients | 668 cases / 668 patients |
| Data leakage prevention | Test patients completely disjoint from specialist training patients (patient-level split) | Patient-level 70/10/20 split with fixed seed; no patient overlap across train/val/test |
| Class imbalance (test) | Varies by task | 10.0% acute / 90.0% non-acute |
| ***Framework*** | | |
| Generalist LLM | GPT-4o | GPT-4o |
| Uncertainty components | $U_{\text{conf}} + U_{\text{conflict}} + U_{\text{coherence}}$ | $U_{\text{conf}} + U_{\text{coherence}}$ (no $U_{\text{conflict}}$ with single specialist) |

ECHO = echocardiography; ECG = electrocardiogram; CXR = chest X-ray; PA = posteroanterior; AP = anteroposterior; IRB = Institutional Review Board; LLM = large language model; $U_{\text{conf}}$ = generation confidence uncertainty; $U_{\text{conflict}}$ = cross-specialist conflict uncertainty; $U_{\text{coherence}}$ = reasoning coherence uncertainty.

## L   Weight Sensitivity Analysis

To validate that the reasoning agent is meaningfully sensitive to weight annotations (Eq. 7), we conduct controlled ablations where the only variable is how weights are presented to the reasoning agent; specialist models and all other components remain identical. Three configurations are evaluated: **Weighted** supplies the specialist-specific weight $w_i = \text{softmax}(\log c_i + \log(1 - \delta_i))$ (Eq. 7) as a prompt-level annotation, where $c_i$ is the generation confidence and $\delta_i$ the conflict score; **No annotation** omits weight fields entirely; **Inverse-weighted** swaps the two specialists' computed weights, keeping all prompt templates and specialist outputs unchanged.

Table 17 reports the cardiovascular results. Correct weight annotations consistently outperform both comparison conditions across all tasks. The inverse-weighted condition yields the largest degradation (average AUROC: 0.798→0.758; average F1: 0.773→0.727), directly confirming that the reasoning agent relies on weight information when forming its synthesis.

For the IU X-Ray setting (Table 18), the system has a single specialist ($A_{\text{CXR}}^w$), so Eq. 7 reduces to $w_i = c_i$; the inverse-weighted condition presents $1 - c_i$, telling the reasoning agent the specialist is uncertain when it is actually confident. The same pattern holds in the single-specialist IU X-Ray setting: providing correct confidence annotations improves performance over omitting them, and inverting the annotation produces the largest drop (F1: 0.537→0.493), consistent with the cardiovascular findings.

## M   Uncertainty Component Ablation

To validate each component of the composite uncertainty score (Eq. 10), we ablate individual terms and measure the impact on routing quality via the Autonomous-to-Intervention F1 Ratio (AIR = $\text{F1}_{\text{auto}}/\text{F1}_{\text{inter}}$), where $\text{F1}_{\text{auto}}$ is the F1 on autonomously handled cases and $\text{F1}_{\text{inter}}$ is the F1 on escalated cases before correction. When one component is removed,

*Table 11.* Inter-patient split of the IU X-Ray dataset used in the acute/non-acute decision task.

| Split | Count | Ratio (%) |
|---|---|---|
| Train | 2,335 | 69.97 |
| Validation | 334 | 10.01 |
| Test | 668 | 20.02 |
| Total | 3,337 | 100.00 |

*Table 12.* Label distribution of the IU X-Ray test split ($n$=668), where `acute` indicates urgent management required and `non-acute` indicates no urgent management required.

| Acuity Label | Count | Ratio (%) |
|---|---|---|
| Non-acute | 601 | 89.97 |
| Acute | 67 | 10.03 |
| Total | 668 | 100.00 |

the uncertainty score is re-normalised over the remaining components using Eq. 10 (e.g., w/o $U_{\text{conf}}$: $U = \lambda_{\text{conflict}} U_{\text{conflict}} + \lambda_{\text{coherence}} U_{\text{coherence}}$, with $\lambda_{\text{conflict}} = \lambda_{\text{coherence}} = \frac{1}{2}$ in our experiment; $U_{\text{conf}}$ only: $U = U_{\text{conf}}$). The adaptive threshold $\theta_P$ is re-calibrated independently for each configuration.

Table 19 reports the cardiovascular results. Removing any single component degrades AIR, with the largest drop from removing $U_{\text{conf}}$ ($-0.290$). However, $U_{\text{conf}}$ alone (AIR=1.270) substantially underperforms the composite metric, confirming that cross-agent conflict and reasoning coherence provide complementary routing information.

For the IU X-Ray setting (Table 20), the system operates with a single specialist, making $U_{\text{conflict}}$ structurally absent; the ablation therefore covers only $U_{\text{conf}}$ and $U_{\text{coherence}}$. Consistent with the cardiovascular results, removing either component degrades AIR, with the larger contribution from $U_{\text{conf}}$ ($-0.241$) relative to $U_{\text{coherence}}$ ($-0.141$), confirming that generation confidence remains the primary routing signal across both settings.

*Table 13.* Step-by-step CXR data preprocessing pipeline.

| Step | Operation | Description | Output |
|------|-----------|-------------|--------|
| 1 | XML parsing | Parse IU X-Ray XML files; extract COMPARISON, INDICATION, FINDINGS, IMPRESSION, and image file references | Structured table (3,955 cases) |
| 2 | Image matching | Match image filenames to PNG files on disk; mutually fill Image1/Image2 if one view is missing | Verified cases |
| 3 | Row filtering | Remove cases with both views missing or with empty / invalid FINDINGS | 3,337 valid cases |
| 4 | Image loading | Load each PNG, convert to RGB, resize to $224{\times}224$, transpose to (3, 224, 224) uint8 | Image tensors |
| 5 | Tokenisation | Lowercase, regex word tokenisation (\b\w+\b) | Token lists |
| 6 | Vocabulary | Build word$\rightarrow$index mapping (min freq $\geq$ 2); special tokens: `<blank>`(0), `<unk>`(1), ``(2), ``(3) | $|\mathcal{V}| \approx 900$ |
| 7 | Index encoding | Convert token lists to index sequences; prepend ``, append ``; truncate to max_len = 100 | Integer sequences |
| 8 | Train/Val/Test split | Stratified patient-level split 70 / 10 / 20 (seed = 42) | Serialised dataset |
| 9 | Image normalisation | Float32, divide by 255 to [0, 1] | Training-ready tensors |

**Note:** Steps 1–3 correspond to data acquisition and quality filtering. Steps 4–7 handle feature engineering (image and text). Steps 8–9 finalise the dataset for model training.

## Orchestrator Agent Prompt

```
prompt = """
You are the Orchestrator Agent in the HetMedAgent system.

## Role & Objective
Your responsibility is to:
(1) parse and enumerate all clinical decision tasks that must be addressed,
(2) activate the appropriate domain specialist agents based on the available modalities,
(3) prepare modality-specific inputs for each activated specialist,
(4) coordinate information flow by collecting and structuring specialist outputs for the downstream
Reasoning Agent.

You are NOT allowed to perform clinical reasoning, diagnosis, risk prediction, or final decision-making.
You only route tasks and organize information.

## Inputs
Patient Context:
{patient_context}

Available Modalities:
{available_modalities}

Clinical Decision Tasks (if provided):
{clinical_decision_tasks}

## Instructions
1. Task Identification
   - Identify and list ALL clinical decision tasks required by the case.
   - If the provided "Clinical Decision Tasks" is incomplete, add missing tasks inferred from the patient
context.
   - Keep task names concise and clinically standard.

2. Agent Activation (Routing)
   - Activate specialist agents strictly based on the available modalities.
   - Only activate agents that can consume the given modalities.
   - If a required modality is missing, keep the task but note that no specialist can be activated for that
modality.

3. Specialist Input Packaging
   - For each activated specialist agent, create an "input" field containing:
     (a) the relevant subset of patient context,
     (b) the specific task(s) that specialist should address,
     (c) the modality data reference/description available in this case.
   - Do not add interpretations or conclusions.

4. Output Structuring for Downstream Reasoning
   - Ensure the output is complete, well-structured, and easy to parse.
   - Do not include any free-form text outside the JSON.

## Output Requirements (Strict)
- Output MUST be valid JSON and MUST follow the schema below exactly.
- Use English only.
- Use standard medical terminology.
- No markdown, no extra commentary, no trailing text.

## JSON Schema
{
  "orchestration": {
    "activated_agents": [
      {
        "agent": "ECHO Specialist",
        "input": "..."
      },
      {
        "agent": "ECG Specialist",
        "input": "..."
      }
    ],
    "tasks": ["..."],
    "patient_context": "...",
    "available_modalities": "..."
  }
}
"""
```

*Figure 8.* Structured prompt $\psi_{task}$ used by the orchestrator agent.

**Clinical Reasoning Agent Prompt**

```
prompt = """
You are the Clinical Reasoning Agent in the HetMedAgent system.

## Role & Objective
Your responsibility is to integrate multimodal diagnostic findings produced by specialist agents and
synthesize them into:
(1) a structured clinical reasoning chain, and
(2) a preliminary decision that directly addresses the specified clinical decision task.

Your output must be evidence-grounded: every key claim should be supported by the provided
integrated evidence and/or standard clinical knowledge/guidelines.

## Inputs
Integrated Evidence (from specialist agents; may include confidence, modality-specific findings, and
notes):
{integrated_evidence}

Clinical Decision Task:
{clinical_decision_task}

## Instructions
Generate a 3-step reasoning chain and a preliminary decision.

Step 1: Key Findings Summary
- Summarize findings from EACH available modality.
- Include relevant patient demographics and clinical context.
- Highlight patient-specific risk factors/comorbidities if present.
- Flag any critical or time-sensitive findings.

Step 2: Clinical Reasoning
- Apply relevant medical knowledge to connect findings to conclusions.
- Reference applicable clinical practice guidelines (cite by name/organization when possible; do not
fabricate).
- Consider differential diagnoses when appropriate.
- Provide risk–benefit considerations if applicable.
- Comment on evidence quality/limitations (e.g., missing modalities, conflicting findings, low-
confidence outputs).

Step 3: Conclusion
- Provide a coherent synthesis that directly answers the clinical decision task.
- Explicitly acknowledge uncertainties if present.

Preliminary Decision
- Provide a clear, actionable preliminary decision statement.

## Output Requirements (Strict)
- Output MUST be valid JSON and MUST follow the schema below exactly.
- Use English only.
- Use standard medical terminology.
- Maintain logical coherence across the three steps.
- No markdown, no extra commentary, no trailing text.

## JSON Schema
{
  "reasoning_chain": {
    "step_1_key_findings_summary": "",
    "step_2_clinical_reasoning": "",
    "step_3_conclusion": ""
  },
  "preliminary_decision": {
    "decision": ""
  }
}
"""
```

*Figure 9.* Structured prompt $\psi_{\mathrm{reason}}$ used by the reasoning agent. The prompt instructs the LLM to reason with clinical knowledge and medical guidelines, and generate preliminary decisions with explicit reasoning chains.

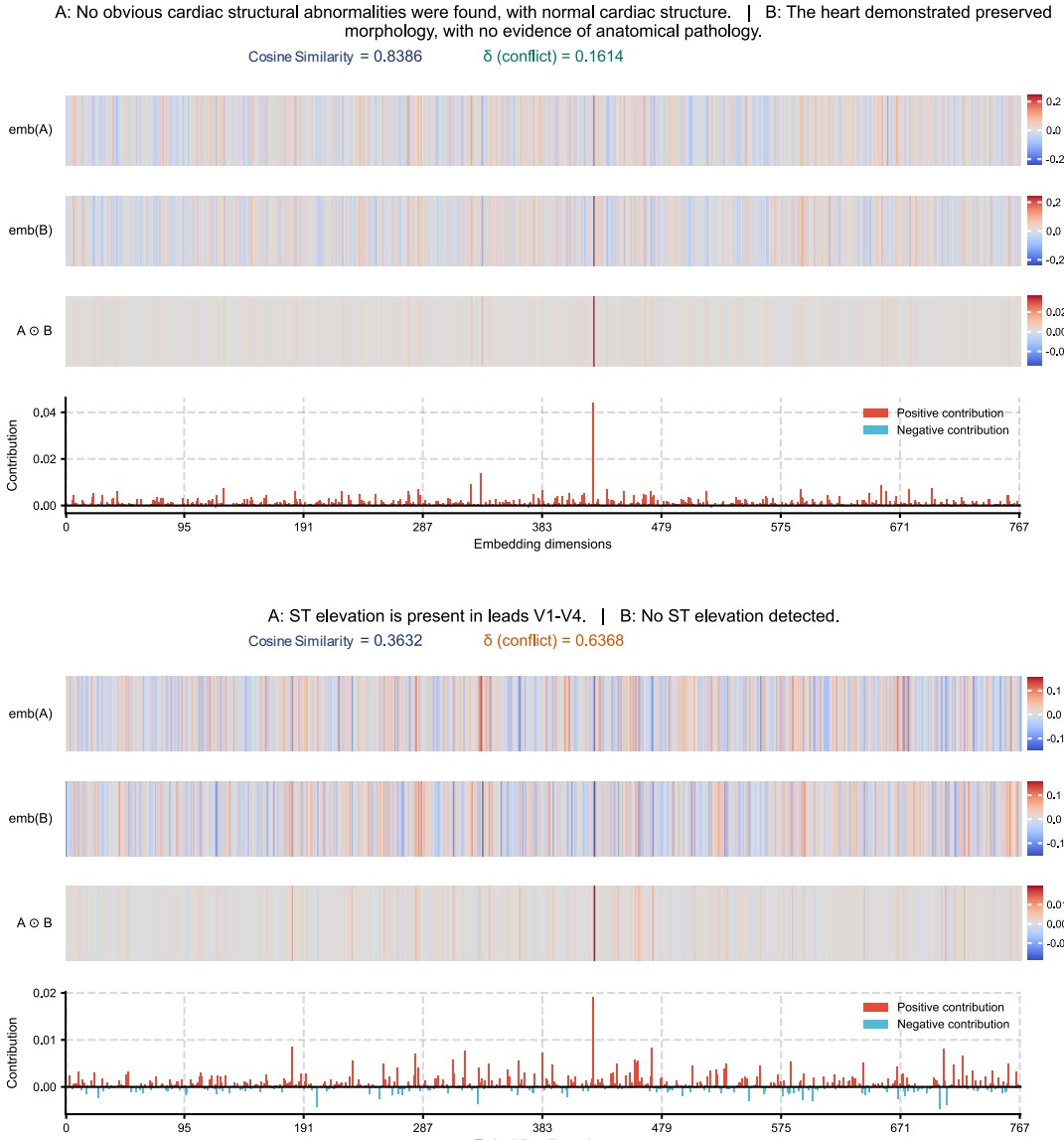

*Figure 10.* Controlled extreme examples for cross-specialist conflict detection. Case (**Top**): semantically equivalent findings expressed with different terminology. Case (**Bottom**): findings with partial lexical overlap but contradictory semantics. *emb*($\cdot$): semantic embedding via PubMedBERT bi-encoder.

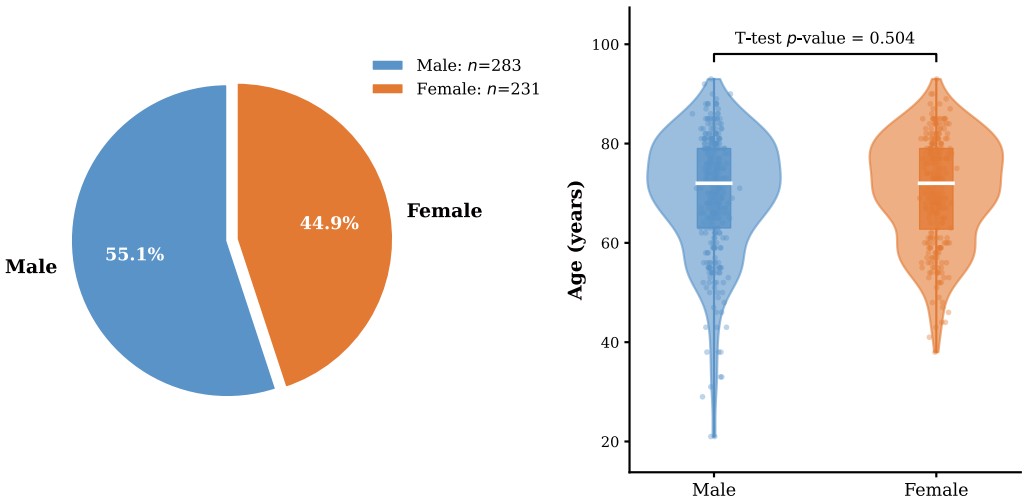

*Figure 11.* Demographic distributions of the multimodal test set. (**Left**) Gender distribution (unique patients). (**Right**) Age distribution by gender (all cases).

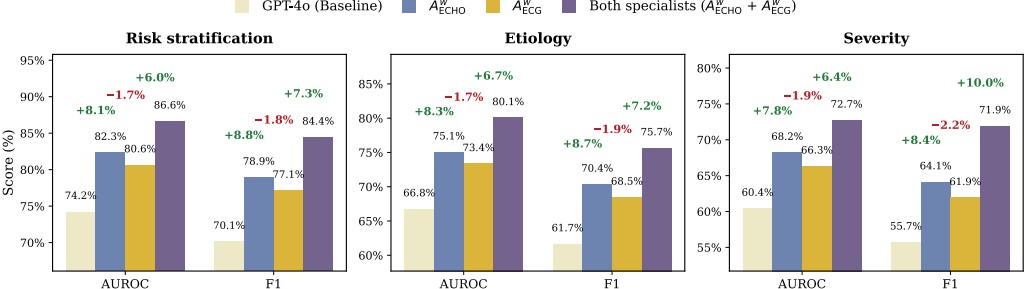

*Figure 12.* Modal ablation experiments for HetMedAgent (w/o Clinician) showing task-level F1 and AUC scores across different modality configurations. ECHO: echocardiography only; ECG: electrocardiogram only; ECHO+ECG: multimodal fusion.

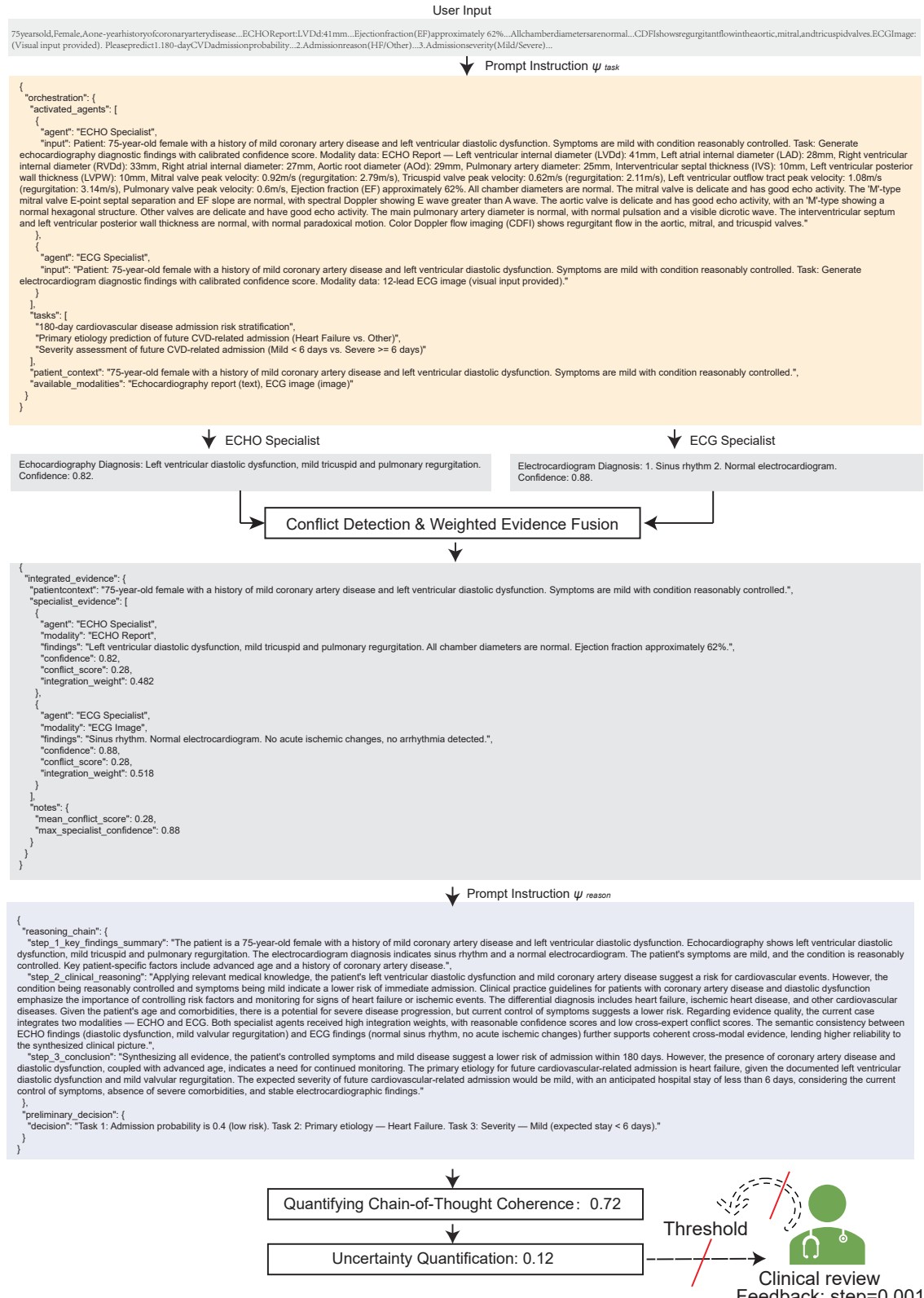

*Figure 13.* Complete routing trace for the representative autonomous case study. The visualization expands Figure 7 by showing the orchestrator dispatch, specialist-agent outputs, conflict-aware evidence fusion, uncertainty-based routing decision, and final autonomous recommendation.

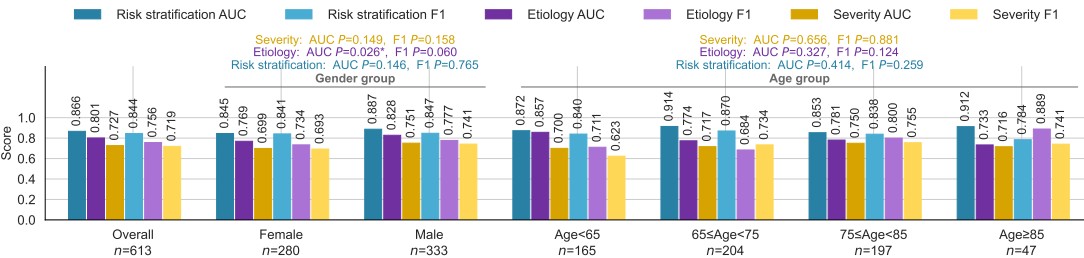

*Figure 14.* Subgroup analysis of HetMedAgent (w/o Clinician) on the cardiovascular clinical test set, stratified by sex (Male/Female) and age group ($<65$, 65–74, 75–84, $\geq 85$). $*p < 0.05$; $**p < 0.01$; $***p < 0.001$; ns: not significant (Fisher's exact test).

*Table 14.* Architectural and training hyperparameters of the $A_{\mathrm{CXR}}^{w}$ specialist model.

| Hyperparameter | $A_{\mathrm{CXR}}^{w}$ |
|---|---|
| ***Architecture*** | |
| Paradigm | Image (dual) $\rightarrow$ Text |
| Encoder type | 4-block CNN ($\times 2$) |
| Decoder type | Transformer |
| Number of layers | 6 |
| Model dimension ($d_{\mathrm{model}}$) | 512 |
| Word embedding dim ($d_{\mathrm{word}}$) | 512 |
| FFN inner dimension ($d_{\mathrm{inner}}$) | 2,048 |
| Number of attention heads | 8 |
| Key/Value dimension ($d_k/d_v$) | 64 / 64 |
| Max sequence length | 100 |
| Input modality / resolution | Image ($224\times224$) $\times 2$ |
| CNN encoder blocks | 4 per view |
| Visual tokens | 98 ($49\times2$) |
| ***Training*** | |
| Optimiser | Adam |
| Learning rate | 1e-4 |
| Adam $\beta_1, \beta_2$ | 0.9, 0.98 |
| $\epsilon$ | $1\times10^{-9}$ |
| Warmup steps | 4,000 |
| Batch size | 8 |
| Dropout rate | 0.1 |
| Label smoothing | 0.1 |
| Max epochs | 100 |
| Early stopping patience | 10 |
| Loss function | CE (ignore PAD) |
| Weight sharing | Embedding–Proj |
| ***Output*** | |
| Output format | Text + confidence |
| Confidence method | Token-prob. agg. |

CE = cross-entropy; PAD = padding token; Embedding–Proj = weight sharing between target word embeddings and output projection layer; Token-prob. agg. = geometric mean of per-token softmax probabilities.

*Table 15.* End-to-end experimental pipeline for the IU X-Ray experiment.

| Stage | Component | Description |
|---|---|---|
| ***Specialist Model*** | | |
| 1 | Specialist training | $A_{\text{CXR}}^w$ is trained end-to-end on the IU X-Ray training split (2,335 samples) to generate radiology findings from paired dual-view CXR images. Best checkpoint selected by validation accuracy (early stopping, patience = 10). |
| 2 | Specialist inference | For each test sample, $A_{\text{CXR}}^w$ receives two CXR views and outputs: (a) a natural-language radiology findings report; (b) a generation confidence score, computed as the geometric mean of per-token softmax probabilities during greedy decoding. |
| ***HetMedAgent Pipeline*** | | |
| 3 | Framework integration | The trained $A_{\text{CXR}}^w$ is plugged into HetMedAgent as a specialist module. The orchestrator agent collects the specialist output (findings + confidence) and the patient's clinical context (INDICATION from XML). |
| 4 | Uncertainty estimation | With a single specialist, the uncertainty estimate uses two of the three components: generation confidence ($U_{\text{conf}}$) and reasoning coherence ($U_{\text{coherence}}$). Cross-specialist conflict ($U_{\text{conflict}}$) is not applicable. |
| 5 | Reasoning & decision | The reasoning agent (GPT-4o) receives the weighted evidence package and applies its structured three-step reasoning chain (evidence synthesis → differential assessment → acuity classification) to produce an `acute` or `non-acute` decision. |
| ***Baseline & Evaluation*** | | |
| 6 | Baseline (ViT-BERT) | End-to-end multimodal classifier: ViT image encoder + BERT text encoder, jointly trained on the same data for direct acuity prediction. Does *not* use the multi-agent pipeline. |
| 7 | Evaluation | Both systems evaluated on the same 668 test cases using Accuracy, Recall, Specificity, Precision, NPV, F1-score, and AUROC. |

NPV = Negative Predictive Value.

*Table 16.* Classification performance on the IU X-Ray acute/non-acute decision task (test set, $n$=668). Best results in **bold**.

| | Accuracy | Recall | Specificity | Precision | NPV | F1-score | AUROC |
|---|---|---|---|---|---|---|---|
| ViT-BERT | 0.837 | 0.716 | 0.850 | 0.348 | 0.964 | 0.468 | 0.783 |
| HetMedAgent (Ours, w/o Clinician) | **0.868** | **0.761** | **0.880** | **0.415** | **0.971** | **0.537** | **0.820** |

**Note:** The ViT-BERT baseline adopts a ViT-based image encoder (ViT-B/16) paired with a BERT-based text encoder for end-to-end acuity classification.

*Table 17.* Weight sensitivity ablation of HetMedAgent (w/o Clinician) on the cardiovascular test set ($n$=613).

| Configuration | Risk Stratification | | Etiology | | Severity | | Average | |
|---|---|---|---|---|---|---|---|---|
| | AUROC | F1 | AUROC | F1 | AUROC | F1 | AUROC | F1 |
| **Weighted (Ours)** | **0.866** | **0.844** | **0.801** | **0.757** | **0.727** | **0.719** | **0.798** | **0.773** |
| No annotation | 0.847 | 0.822 | 0.779 | 0.732 | 0.704 | 0.693 | 0.777 | 0.749 |
| Inverse-weighted | 0.783 | 0.757 | 0.766 | 0.718 | 0.725 | 0.706 | 0.758 | 0.727 |

**Note:** "Weighted" computes each specialist's weight via Eq. 7: $w_i = \text{softmax}(\log c_i + \log(1 - \delta_i))$, where $c_i$ is the specialist's generation confidence and $\delta_i$ its conflict score. "No annotation" omits weight fields entirely. "Inverse-weighted" swaps the two specialists' computed weights, keeping all prompt templates and specialist outputs unchanged.

*Table 18.* Weight sensitivity ablation of HetMedAgent (w/o Clinician) on the IU X-Ray test set ($n$=668).

| Configuration | Manipulation | AUROC | F1 |
|---|---|---|---|
| **Weighted (Ours)** | Correct $c_i$ in prompt | **0.820** | **0.537** |
| No annotation | Confidence field removed | 0.805 | 0.514 |
| Inverse-weighted | $1 - c_i$ replaces $c_i$ | 0.791 | 0.493 |

**Note:** With a single specialist ($A_{\text{CXR}}^w$), there is no inter-specialist conflict ($\delta_i = 0$), so the weight annotation reduces to the generation confidence $c_i$ (Eq. 7). "Inverse-weighted" presents $1 - c_i$, telling the reasoning agent the specialist is uncertain when it is actually confident (and vice versa).

*Table 19.* Uncertainty-component ablation on the cardiovascular test set ($n$=613).

| Configuration | Escalated | Risk Stratification | | | Etiology | | | Severity | | | Average |
| | | $F1_{\text{auto}}$ | $F1_{\text{inter}}$ | AIR↑ | $F1_{\text{auto}}$ | $F1_{\text{inter}}$ | AIR↑ | $F1_{\text{auto}}$ | $F1_{\text{inter}}$ | AIR↑ | AIR↑ |
|---|---|---|---|---|---|---|---|---|---|---|---|
| Full (3 components) | 97 (15.8%) | 0.890 | 0.543 | 1.639 | 0.818 | 0.479 | 1.708 | 0.783 | 0.463 | 1.691 | **1.679** |
| w/o $U_{\text{conf}}$ | 76 (12.4%) | 0.822 | 0.608 | 1.352 | 0.789 | 0.554 | 1.424 | 0.705 | 0.507 | 1.391 | 1.389 |
| w/o $U_{\text{conflict}}$ | 81 (13.2%) | 0.850 | 0.598 | 1.421 | 0.802 | 0.534 | 1.502 | 0.737 | 0.506 | 1.457 | 1.460 |
| w/o $U_{\text{coherence}}$ | 84 (13.7%) | 0.876 | 0.585 | 1.497 | 0.809 | 0.516 | 1.568 | 0.753 | 0.491 | 1.534 | 1.533 |
| $U_{\text{conf}}$ only | 63 (10.3%) | 0.763 | 0.619 | 1.233 | 0.739 | 0.566 | 1.306 | 0.680 | 0.535 | 1.271 | 1.270 |
| No routing | 0 (0%) | 0.844* | — | — | 0.757* | — | — | 0.719* | — | — | — |

**Note:** F1 over all cases with no escalation. When one component is removed, the uncertainty score is re-normalised over the remaining components using Eq. 10: e.g., w/o $U_{\text{conf}}$: $U = \lambda_{\text{conflict}} U_{\text{conflict}} + \lambda_{\text{coherence}} U_{\text{coherence}}$, with $\lambda_{\text{conflict}} = \lambda_{\text{coherence}} = \frac{1}{2}$ in our experiment; $U_{\text{conf}}$ only: $U = U_{\text{conf}}$. The adaptive threshold $\theta_P$ is initialized to 0.5 and re-calibrated independently for each configuration.

*Table 20.* Uncertainty-component ablation on the IU X-Ray test set ($n$=668).

| Configuration | Escalated | $F1_{\text{auto}}$ | $F1_{\text{inter}}$ | AIR↑ |
|---|---|---|---|---|
| Full (2 components) | 85 (12.7%) | 0.573 | 0.384 | **1.492** |
| w/o $U_{\text{conf}}$ | 56 (8.4%) | 0.549 | 0.439 | 1.251 |
| w/o $U_{\text{coherence}}$ | 67 (10.0%) | 0.558 | 0.413 | 1.351 |
| No routing | 0 (0%) | 0.537* | — | — |

**Note:** F1 over all cases with no escalation. With a single specialist, $U_{\text{conflict}}$ is structurally absent and the uncertainty score reduces to $U = \lambda_{\text{conf}} U_{\text{conf}} + \lambda_{\text{coherence}} U_{\text{coherence}}$, with $\lambda_{\text{conf}} = \lambda_{\text{coherence}} = \frac{1}{2}$ in our experiment. When one component is removed, the score reduces to the remaining component alone: e.g., w/o $U_{\text{conf}}$: $U = U_{\text{coherence}}$; w/o $U_{\text{coherence}}$: $U = U_{\text{conf}}$. The adaptive threshold $\theta_P$ is initialized to 0.5 and re-calibrated independently for each configuration.

