# OpenReview forum: "Why Specialist Models Still Matter: A Heterogeneous Multi-Agent Paradigm for Medical Artificial Intelligence"
_ICML.cc/2026/Conference — ICML 2026 regular_

### Official Review · Reviewer_sFs3 · 2026-03-06

**Soundness:** 3
**Presentation:** 3
**Significance:** 3
**Originality:** 3
**Overall Recommendation:** 5
**Confidence:** 4

**Summary:**

This work proposes a heterogeneous multi-agent framework to tackle hallucinations and lack of domain-specific precision of generalist LLMs in clinical settings. The paper focuses on collaborative architecture that orchestrates generalist LLMs as reasoning engines, specialist models for modality specific analysis, and human clinicians as final judges. The authors leverage an uncertainty quantification approach that assesses confidence across modality-specific findings and reasoning coherence to flag clinician review. Furthermore, the framework weights specialist outputs and a threshold calibration system that learns from clinician agent feedback to enhance safety. Authors demonstrate that on cardiovascular tasks, this collaborative approach outperforms standalone medical LLMs and existing multi-agent systems.

**Compliance With Llm Reviewing Policy:**

Affirmed.

**Final Justification:**

The authors adequately addressed my concerns about generalizability and clarified the conflict detection mechanism. The author's defense of using natural language as the interface is pragmatic. Overall, assuming the author's proposed edits, the rebuttal strengthens the paper and I have raised my score to Accept.

**Key Questions For Authors:**

1. The current approach adjusts the intervention threshold based on clinician feedback, which is simulated using retrospective ground truth data. How do the authors anticipate clinician interaction with the system in a live setting, specifically regarding the cognitive load and time of reviewing reasoning?
2. Relatedly, how do the authors anticipate the threshold calibration dynamics will change when relying on real, potentially noisy, human feedback?
3. Can the authors discuss or quantify the information loss that occurs when requiring specialist models to output natural language summaries? How can this framework adapt to incorporate continuous embedding rather than text?
4. Regarding weakness 3, how would the system prevent contradictory clinical statements sharing similar, dense terminology from yielding high cosine similarity?

**Limitations:**

The paper can be improved with a conclusion section to discuss the technical and methodological limitations including computational generalizability to other complex medical domains, the human-machine interaction friction, and other practical deployment challenges.

**Strengths And Weaknesses:**

Strengths:

1. The work provides evidence that heterogeneous multi-agent frameworks can potentially outperform generalist medical foundation models which are often development resource intensive. The collaborative approach with human clinician review enables modular approach to medical AI deployment.
2. The design of the uncertainty-based routing mechanism is a novel contribution. The work introduces a dynamic approach to human in the loop decision-making.
3. The use of the reasoning chain combined with the evidence weighting makes the system more interpretable for human experts.
4. The baselines used for comparisons are strong, including medical-specific models and multi-agent systems. The ablation studies are thorough.

Weaknesses:

1. The framework is assessed on only the cardiovascular specialty and only two modalities. While the framework is theoretically modular, confirming its generalizability requires testing on clinical domains with different data types.
2. The specialist models are required to convert their modalities into natural language outputs before passing the findings to the orchestrator. This may reduce complex, high dimensional data into text, potentially removing nuanced structural or spatial features that a native multimodal LLM might preserve.
3. Cosine similarity is used to calculate the conflict between specialist model outputs. This might be insufficient to capture clinical contradictions that are semantically similar.
4. Since the framework relies on human review, the experiments evaluating the adaptive threshold seem to rely on retrospective data rather than a prospective real-time study with human experts.
5. The paper ends abruptly. A dedicated conclusion section synthesizing the technical and practical impact of this work can strengthen the paper.

---

> ### Author Rebuttal · Authors · 2026-03-31
>
> We thank Reviewer sFs3 for the careful reading and positive assessment of
> our collaborative architecture, routing mechanism, and interpretability.
>
> **W1 -- Generalizability.**
> We conducted a supplementary validation on the public IU X-Ray dataset,
> whose specialty, imaging type, and report format differ substantially from
> our cardiovascular tasks. We added a chest X-ray–radiology report
> specialist and evaluated on an acute/non-acute decision-making task against
> existing multimodal models. Results (see https://anonymous.4open.science/r/ICML_rebutal-0B83/1Supplementary%20Scenario.PNG) confirm cross-domain
> generalization. Setup and analysis will be added to the revision.
>
> **W2&Q3 -- Text Interface and Information Loss.**
> Natural language was chosen as the inter-agent interface because (i) it
> provides a unified protocol across modalities without shared-representation
> alignment; (ii) human-readable intermediate outputs support traceability
> and clinician oversight; and (iii) it offers a simple, robust interface for
> the LLM-based reasoning agent. Despite text-only communication,
> specialist-augmented performance substantially exceeds both multimodal
> LLM-only and multi-agent baselines, indicating that specialist outputs
> retain clinically relevant information.
>
> As a future extension, we propose a hybrid mechanism where each specialist
> produces both a structured textual summary (for interpretability) and
> continuous embeddings (for multimodal fusion). The reasoning module could
> then jointly consume both representations. We will discuss this in the
> revision.
>
> **W3&Q4 -- Conflict Detection.**
> Given specialist outputs $F_i$ and $F_j$, we encode them with a
> PubMedBERT-based biomedical bi-encoder into normalized embeddings $e_i$,
> $e_j$ and compute $\delta_{ij}=1-\cos(e_i,e_j)$, capturing semantic
> inconsistency rather than lexical overlap. To handle modality-specific
> terminology biases, we apply a shared-direction adjustment
> $\tilde{e}=\mathrm{norm}(e+\alpha\mu)$, where $\boldsymbol{\mu}$ is estimated from specialist outputs and $\alpha$
> is a scaling hyperparameter. Controlled examples (see https://anonymous.4open.science/r/ICML_rebutal-0B83/3Conflict%20Cases.png)
> confirm the mechanism distinguishes terminology variation from genuine
> conflict. Details will be added to the revision.
>
> **W4&Q2 -- Retrospective Evaluation and Noisy Feedback.**
> The current threshold calibration uses retrospective ground truth to
> simulate clinician feedback, enabling controlled assessment of convergence
> properties. Prospective studies require IRB approval and deployment
> infrastructure beyond the current scope.
>
> With real feedback, two primary noise sources arise: inter-clinician
> variability on borderline cases and occasional annotation errors. The
> threshold update (Equation 12) uses an exponential moving average providing
> inherent temporal smoothing—a single erroneous correction is attenuated
> by accumulated prior feedback. For added robustness, we plan
> multi-clinician consensus before updates, exclusion of cases with highly
> divergent responses, and bounded update magnitudes to prevent threshold
> jumps from outliers. Prospective validation is an essential next step; we
> will discuss this limitation and the planned mitigation strategies in the
> revision.
>
> **Q1 -- Cognitive Load and Review Time.**
> When a case is escalated, clinicians receive a structured package:
> three-step reasoning chain, decomposed uncertainty sources, and specialist
> evidence with confidence scores—enabling focused review rather than
> re-analysis of raw data. We plan an interactive visualization dashboard and
> formal usability studies (e.g., NASA-TLX) to quantify cognitive load under
> real conditions. These deployment considerations will be discussed in the
> revision.
>
> **W5 -- Conclusion Section.**
> A dedicated Conclusion will synthesize the contribution, discuss
> limitations (text-interface bottleneck, threshold calibration under noise,
> deployment challenges), and outline future directions.

---

> > ### Author Rebuttal · Reviewer_sFs3 · 2026-04-01
> >
> > I thank the authors for the detailed rebuttal, particularly with the validation on the X-Ray dataset to address generalizability. Assuming the new experiments, conflict detection explanation, and a dedicated conclusion/limitations section are integrated into the revised version, I am happy to raise my score to an Accept.

---

> > > ### Author Response · Authors · 2026-04-02
> > >
> > > We sincerely thank Reviewer sFs3 for the constructive and thorough review, as well as for acknowledging our rebuttal. Your feedback has meaningfully improved the quality of this work.

---

### Official Review · Reviewer_BJ3A · 2026-03-08

**Soundness:** 2
**Presentation:** 3
**Significance:** 3
**Originality:** 2
**Overall Recommendation:** 2
**Confidence:** 4

**Summary:**

The paper introduces HetMedAgent, a multi-agent framework which consists of generalist and specialist agents to coordinate and execute tasks with confidence scores. The paper provides a uncertainty-aware routing mechanism to notify whenever clinician intervention is required. The authors compare their method to existing medical LLMs and agent systems on a cardiovascular dataset with three downstream tasks, and show that HetMedAgent consistently outperforms these baselines. The authors also provide ablations on each component of the agentic system to show their necessity.

**Compliance With Llm Reviewing Policy:**

Affirmed.

**Final Justification:**

I stick with my reject claim. While the authors tried to solve some of my queries: I still believe the confidence scoring is misleading, the optimization objective to prompting is wrong. It is also weird that the confidence via LLM probabilities help clinical-decision making as LLMs have been shown to confidently hallucinate in such tasks. It might hint at model learning spurious correlations than actually reasoning. The scope of experimentation, while proposing multi-agent, is stuck to 2-3 modalities which might not be sufficient for clinical decision-making.

**Key Questions For Authors:**

- While the design is interesting, it is only tested with 2 modalities at hand. Real clinical decision-making involves much more evidence integration. I find it unclear if the proposed mechanisms scale beyond 2 relatively aligned cardiovascular modalities.
- While $c_i$ is a confidence score, the main paper refers to it coming via "ensemble methods or calibration techniques". However, the appendix states $c_i$ is the mean of probabilities of the generated text - which refers to how fluent the LM might be, not the correctness of the medical statment. Wrong answers might also end up getting high confidences, and this now depends on how the LM is configured
- In S3.2.4, doesn't medical terminology also affect the similarity? If two modalities have different terms used to describe the same reasoning, that would cause a problem I feel due to embeddings not being so similar (moreover since the specialists are trained independently)
- In eq8, the fusion is a weighted aggregation but I am confused how this is actually implemented. If we give the findings and weights to the generalist LLM, it is actually not computing and optimizing this objective? Moreover, the fusion is confusing, shouldn't the fusion be multiplicative of the likelihoods?  $P(D | F_1, \dots, F_k) \propto P(D) \prod_i P(F_i | D)$ since the specialists are independent?
- The downstream evaluation is quite limited - its tested only on one cohort on one clinical setting which makes it difficult to understand if the performance gains are dataset specific or from the agentic architecture.

Minor:
- How are the specialist model's architectures chosen?

I look forward to have a discussion with the authors on the paper. However, in the current state without any examples of conversation, or confidences, and the above mentioned points, I lean towards a reject.

**Limitations:**

Mentioned along with Key Questions.

**Strengths And Weaknesses:**

- Decomposing the workflow into modality-specific specialist agents makes sense, and aligns somewhat with real world workflows. Moreover, this makes the system also modular and also allows tuning of the specialists individually
- The paper tackles a crucial task of alerting clinicians when the model confidence is low. This is quite important when such systems are deployed in real-world, since the tasks are high-stake.

---

> ### Author Rebuttal · Authors · 2026-03-31
>
> We appreciate Reviewer BJ3A's interest in discussing the work further.
> Below we respond point by point and will revise the manuscript
> accordingly.
>
> **Q1&Q5 -- Scalability and Single-Cohort Evaluation.**
> Beyond the two core examination modalities (ECHO reports and ECG images),
> HetMedAgent incorporates baseline clinical information as contextual input.
> ECHO and ECG were chosen as the initial setting because they are routinely
> co-interpreted in cardiovascular practice.
>
> To demonstrate broader scalability, we conducted experiments on the publicly
> available IU X-Ray dataset, which differs in clinical specialty, imaging
> type, and report format. We integrated a "chest X-ray--radiology report"
> specialist into HetMedAgent and performed an acute/non-acute decision-making task,
> comparing against existing multimodal models. Results (see https://anonymous.4open.science/r/ICML_rebutal-0B83/1Supplementary%20Scenario.PNG) confirm
> effective transfer to this different scenario. Full details will appear in
> the revision.
>
> **Q2 -- Meaning and Role of Confidence Scores.**
> We apologize for the ambiguity. The specialist confidence score aggregates
> token-level generation probabilities; it reflects generation confidence,
> not probability of clinical correctness. Incorrect conclusions can receive
> high confidence---a known limitation of generative models. We treat this
> score as an internal uncertainty signal: lower confidence indicates less
> stable generation and retains warning value.
>
> Crucially, confidence does not determine routing alone. It is combined with
> cross-specialist conflict and reasoning incoherence into the composite
> uncertainty (Equation 10). High-confidence specialist output still triggers
> clinician review if substantial conflict or reasoning inconsistency is
> detected. Clinician feedback then calibrates the review threshold via
> Equation 12. We have also added a case study (see https://anonymous.4open.science/r/ICML_rebutal-0B83/4Reasoning%20Case.png) illustrating the full pipeline for a representative patient, including orchestration, specialist outputs with confidence scores, conflict detection and evidence fusion, the three-step reasoning chain, and threshold-based routing.
>
> **Q3 -- Medical Terminology and Conflict Detection.**
> Conflict is measured in a medical semantic embedding space, not at the
> lexical level. Given specialist outputs $F_i$ and $F_j$, we encode them
> with a PubMedBERT-based biomedical bi-encoder into normalized embeddings
> $\mathbf{e}_i$, $\mathbf{e}_j$, compute
> $s_{ij}=\cos(\mathbf{e}_i,\mathbf{e}_j)$, and define
> $\delta_{ij}=1-s_{ij}$.
>
> To handle consistent phrasing biases across modalities, we apply a
> shared-direction adjustment:
> $\tilde{\mathbf{e}}=\mathrm{norm}(\mathbf{e}+\alpha\boldsymbol{\mu})$,
> where $\boldsymbol{\mu}$ is estimated from specialist outputs and $\alpha$
> is a scaling hyperparameter. Controlled examples examining semantically
> equivalent findings with different terminology, and lexically overlapping
> but contradictory findings (see https://anonymous.4open.science/r/ICML_rebutal-0B83/3Conflict%20Cases.png), confirm that the mechanism
> distinguishes terminology variation from genuine conflict. Details will be
> added to the revision.
>
> **Q4 -- Fusion Formula (Equation 8).**
> We apologize that the original text gave an overly strong probabilistic impression. Equation 8
> formalizes weighted evidence aggregation at a conceptual level; the LLM
> does not explicitly optimize it as a training objective. Each specialist
> generates findings and a confidence score, the system computes weights
> incorporating confidence and conflict scores, and the resulting $(F_i, w_i)$ pairs are
> provided as structured input to the reasoning agent.
>
> A product-of-experts formulation would require calibrated probability
> distributions over a shared label space under conditional independence.
> Our specialists produce natural-language findings across different
> modalities, so these assumptions do not hold. Moreover, cross-modality
> evidence is often complementary rather than redundant---e.g., ECHO
> captures structural/functional abnormalities while ECG reflects electrical
> conduction---and reliable diagnosis frequently requires integrating both.
> Weighted aggregation accommodates such complementarity by combining
> evidence in proportion to estimated reliability, whereas a multiplicative
> formulation would require each source to independently support the same
> conclusion. We will clarify this rationale and temper the probabilistic
> interpretation of Equation 8.
>
> **Q6 -- Specialist Architecture Choice.**
> Architectures were selected by input modality and the need for
> standardized text output. ECHO reports (text) use a Transformer
> encoder-decoder with an LSTM session layer for cross-section context; ECG
> images use a CNN encoder with a Transformer decoder for diagnostic text
> generation. These are modality-compatible choices producing interpretable
> outputs and confidence signals, not claimed as uniquely optimal. A
> clarification will be added.

---

> > ### Author Rebuttal · Reviewer_BJ3A · 2026-04-04
> >
> > I thank the authors for the rebuttal.
> > (1) My concern regarding single cohort is slightly solved -- I know the character the limit of the rebuttal was short. Can the authors explain a bit on the modalities + the experiment here?
> > (2) Token-level likelihood is still not a calibrate measure of correctness, and the I feel the fact that it influences the clinical decision is not so correct (might be miscalibrated). Have the authors tested what happens when this uncertainty is removed?
> > (3) If I understand, eq 8 is not an optimization or aggregation where the LLM just answers based on the values provided. I think there needs to a major rewrite to this part, as it is extremely misguiding to the reader that the optimization objective is turned to prompting. Since the framework lacks explicit likelihoods, additive formulation is also not probabistically correct.

---

> > > ### Author Response · Authors · 2026-04-05
> > >
> > > We sincerely thank Reviewer BJ3A for the continued engagement and for the opportunity to elaborate on the supplementary experiment.
> > >
> > > **Q1:** Below we provide a comprehensive, table-driven description of the modalities, the specialist model architecture, the dataset, and the full experimental pipeline.
> > > Table R1 (https://anonymous.4open.science/r/ICML_r-85ED/Table%20R1.PNG) provides a structured, side-by-side comparison between the original cardiovascular setting and the new chest radiology setting across key dimensions.
> > > Table R2 (https://anonymous.4open.science/r/ICML_r-85ED/Table%20R2.PNG) presents a side-by-side comparison of all specialist architectures and their hyperparameters.
> > > Table R3 (https://anonymous.4open.science/r/ICML_r-85ED/Table%20R3.PNG) details each step of the data preprocessing pipeline for the IU X-Ray experiment.
> > > Table R4 (https://anonymous.4open.science/r/ICML_r-85ED/Table%20R4.PNG) details the end-to-end pipeline from specialist training to final evaluation.
> > >
> > > **Q2:** We agree that token-level confidence is not well-calibrated for clinical correctness. To test its contribution, we ablate each uncertainty component from the routing score (Eq. 10).
> > > Tables R5 (https://anonymous.4open.science/r/ICML_r-85ED/Table%20R5.PNG) and Table R6 (https://anonymous.4open.science/r/ICML_r-85ED/Table%20R6.PNG) show that removing $U\_{\text{conf}}$ reduces AIR (e.g., Table R5: Avg AIR 1.679→1.389; Table R6: 1.492→1.251), indicating that token-level confidence is an impactful component in the current routing mechanism. That said, $U\_{\text{conf}}$ alone (R5: AIR=1.270) still underperforms the composite metric, confirming that cross-agent conflict and reasoning coherence provide complementary information. We acknowledge the reviewer's point that the calibration quality of token-level confidence itself still has room for improvement. Enhancing per-modality confidence calibration (e.g. Platt scaling) is an important direction for future work, and we will discuss this explicitly in the revised manuscript.
> > >
> > > **Q3:** We fully agree that Equation 6 conflates a formal aggregation objective with what is implemented as structured prompting. In the revision, we will remove all probabilistic language and rewrite Equation 6 as a deterministic evidence assembly protocol:
> > >
> > > $$\text{Input}\_{\text{reason}} = \text{Context}(V\_C) \oplus \bigoplus\_{i=1}^{k} \left\langle F\_i^w,\, w\_i \right\rangle$$
> > >
> > > where $\oplus$ denotes string concatenation and $w\_i$ are per-case weights computed from specialist confidence and conflict scores via Eq. 7 (not likelihoods). Correspondingly, Equation 8 is simplified to:
> > >
> > > $$D\_{\text{prelim}} = \text{LLM}\_{A^r}\left(\psi\_{\text{reason}}, \text{Input}\_{\text{reason}}\right)$$
> > >
> > > directly referencing the assembled input defined in Equation 6. To empirically validate that the weight values carry genuine information despite being implemented via prompting, Table R7 (https://anonymous.4open.science/r/ICML_r-85ED/Table%20R7.png) presents a controlled ablation in which the *only* variable is how weights are presented: the 4.0% AUROC gap between correct and inverted weights confirms the LLM is meaningfully sensitive to these annotations, and inverted weights perform worse than no weights (−1.9% AUROC), showing incorrect annotations actively mislead the reasoning agent. Table R8 (https://anonymous.4open.science/r/ICML_r-85ED/Table%20R8.png) replicates this ablation in the IU X-Ray setting (single specialist): the 2.9% AUROC gap between correct and inverted confidence annotations confirms the finding generalises across clinical domains.
> > >
> > > We hope this detailed, table-driven explanation fully resolves Reviewer BJ3A's remaining concerns. All of the above will be incorporated into the revised manuscript.

---

### Official Review · Reviewer_zJNm · 2026-03-12

**Soundness:** 3
**Presentation:** 3
**Significance:** 3
**Originality:** 4
**Overall Recommendation:** 5
**Confidence:** 5

**Summary:**

This paper introduces HetMedAgent, a collaborative framework that combines the reasoning capabilities of generalist Large Language Models (LLMs) with the precision of domain-specific specialist models and human oversight.
Its key contributions are:
1) Heterogeneous Architecture: It orchestrates a "team" where generalist LLMs manage tasks and specialist agents analyze specific modalities (like ECGs or echocardiograms), avoiding the pitfalls of using a single monolithic model for everything.
2) Uncertainty-Based Routing: The system calculates an uncertainty score based on confidence gaps and conflicts between agents. If uncertainty is too high, the decision is automatically escalated to a human clinician.
3) Adaptive Calibration: The framework learns from human feedback, dynamically adjusting its intervention thresholds to balance efficiency with patient safety.
Experiments on cardiovascular datasets showed that this multi-agent approach significantly outperformed both standalone medical LLMs and other multi-agent systems.

**Compliance With Llm Reviewing Policy:**

Affirmed.

**Key Questions For Authors:**

1) The adaptive threshold calibration experiment simulates clinician feedback using ground-truth labels. Have you conducted (or do you plan) any studies with real clinicians in the loop to validate that the uncertainty signals, escalation decisions, and adaptation dynamics behave as intended under noisy, heterogeneous human feedback?
---Evidence that the system remains stable and beneficial under real clinician interaction would significantly strengthen our assessment of soundness and practical significance; if calibration is fragile to noisy feedback, that would be an important limitation to highlight.

2) The current evaluation is based on 613 retrospective cardiovascular cases from one institution. How confident are you that the same architecture and uncertainty-based routing will generalize to other hospitals, populations, and modalities (e.g., radiology, pathology)? Are there any preliminary results or constraints you can share?
---Demonstrating or carefully bounding generalization would increase the perceived breadth of impact; if generalization is currently unknown, we would view the contribution as a strong but domain-specific proof-of-concept.

3) Given the high-stakes medical setting, have you analyzed performance and escalation behavior across patient subgroups (e.g., age, sex, comorbidities) or considered governance aspects such as responsibility allocation when the system’s recommendation conflicts with the clinician’s judgment?
---Additional analysis of fairness and governance would bolster the significance and responsible-ML aspects; if such analyses are out of scope, explicitly stating them as limitations would help set appropriate expectations for deployment.

**Limitations:**

Yes. The authors acknowledge several important limitations, including the use of a relatively small, single-institution retrospective dataset and the reliance on simulated clinician feedback for adaptive calibration rather than real-world human interaction. It would further strengthen the paper to more explicitly discuss (i) how robust the uncertainty-based routing and conflict-aware fusion are under distribution shifts and noisy clinician behavior in practice, (ii) the extent to which the current cardiovascular-focused evaluation supports claims about broader "medical artificial intelligence", and (iii) potential fairness, regulatory, and governance issues that arise when delegating parts of clinical decision-making to a heterogeneous multi-agent system.

**Strengths And Weaknesses:**

Soundness
Strengths:
Robust Experimental Design: The authors compare their framework against a comprehensive set of baselines, including state-of-the-art medical LLMs (e.g., Meditron, PMC-LLaMA) and other multi-agent frameworks (e.g., MedAgents, AgentClinic). This benchmarking validates the claim that the heterogeneous approach outperforms monolithic models.
Rigorous Methodology: The proposed mechanisms—specifically "conflict-aware evidence fusion" and multidimensional "uncertainty-based routing" (combining confidence, conflict, and coherence)—are mathematically grounded and logically justified for the problem of integrating noisy multimodal data.
Ablation Studies: The paper includes detailed ablation studies (analyzing different LLM backends, modality combinations, and fusion strategies), which isolate the contributions of specific components and strengthen the technical claims.
Weaknesses:
Validation of Human Interaction: In the "Adaptive Threshold Calibration" experiment, the "clinician feedback" was simulated using ground truth labels rather than actual human interaction. While this validates the mathematical adaptation algorithm, it does not account for the noise or inconsistency of real-world human decision-making.
Dataset Scope: The evaluation relies on a retrospective dataset of 613 cardiovascular cases from a single institution. While sufficient for a proof-of-concept, claims regarding broad applicability to "medical artificial intelligence" would benefit from larger, multi-institutional, or cross-domain datasets (e.g., including radiology or pathology beyond cardiac data).

Presentation
The paper is well-presented, structured, and easy to follow.
Strengths:
Clarity and Structure: The narrative logically flows from the limitations of generalist LLMs to the proposed multi-agent solution. The breakdown of agent roles (Orchestrator, Specialist, Reasoning, Clinician) is intuitive.
Visual Aids: Figures 1, 2, and 3 clearly illustrate the complex architecture and decision-making workflow, making the system easy to conceptualize.
Reproducibility: The inclusion of specific prompt templates (Figures 9 and 10 in the appendix) and detailed architectural hyperparameters for the specialist models significantly enhances reproducibility.
Weaknesses:
Specialist Model Details: While the architecture of the specialist models is described in the appendix, the main text focuses heavily on the orchestration. A brief mention of how the specialist models are trained or sourced in the main methodology section would improve immediate understanding of the "heterogeneous" nature of the system.

Significance
The paper addresses a highly relevant and timely problem in medical AI: the "generalist vs. specialist" dilemma.
Strengths:
Practical Relevance: By addressing the specific limitations of generalist LLMs (hallucination, inability to process raw medical signals), the paper proposes a pragmatic path forward for deploying AI in high-stakes healthcare environments.
Safety Focus: The emphasis on "uncertainty-based routing" and keeping the "clinician in the loop" is significant. It aligns technical innovation with the ethical and safety requirements of real-world medicine, moving beyond pure performance metrics to actionable deployment strategies.
Paradigm Shift: The argument for orchestrating specialized agents rather than forcing all knowledge into a single foundation model is a significant conceptual contribution that could influence future system designs in medical AI.

Originality
The work demonstrates a high degree of originality, particularly in its architectural synthesis and control mechanisms.
Strengths:
Novel Architecture: While multi-agent systems exist, the HetMedAgent framework's specific combination of generalist LLMs (for reasoning/routing) with trained domain-specific transformer specialists (for raw data analysis) is a novel and effective design pattern.
Innovative Routing Mechanism: The introduction of a multidimensional uncertainty metric that specifically measures "cross-agent conflict" and "reasoning coherence" to trigger human intervention is a creative and rigorous contribution.
Adaptive Calibration: The "adaptive threshold calibration" mechanism, which dynamically adjusts the autonomy level of the system based on historical feedback, is a novel addition that adds temporal intelligence to the system, allowing it to "learn" the appropriate level of caution over time.

---

> ### Author Rebuttal · Authors · 2026-03-31
>
> We thank Reviewer zJNm for the positive assessment of our framework's
> modular design and clinical applicability, and for the constructive
> suggestions on robustness, generalizability, and governance.
>
> **Q1 -- Noise Robustness and Real Clinician Feedback.**
> Ground-truth labels provide a clean calibration signal, whereas real
> clinicians introduce variability from fatigue, specialty bias, and
> individual judgment differences. The small step size in Equation 12 (0.001)
> offers some baseline noise damping. In Section 5 we will describe a planned
> momentum-based extension,
> $\theta_P \leftarrow (1-\alpha)\theta_P + \alpha\,\theta_P^{\text{new}}$,
> to further smooth out single-instance outliers. We will also state
> explicitly that a prospective study with real clinician
> interaction—including measurement of inter-clinician variability and its
> effect on threshold stability—is needed before any deployment claim can
> be made. If calibration turns out to be fragile under noisy feedback, we
> agree this should be highlighted as a key limitation.
>
> **Q2 -- Generalizability Across Institutions and Modalities.**
> We agree that the single-specialty cardiovascular evaluation, while
> representative of a common clinical workflow, is insufficient to fully
> demonstrate the framework's scalability across broader clinical settings.
>
> We first clarify that, in addition to ECHO reports and ECG images as the
> two core examination modalities, HetMedAgent incorporates baseline clinical
> information as contextual input during the decision process. The
> current experimental scope therefore involves joint modeling of
> examination-modality evidence and clinical background information. ECHO and
> ECG were selected as the initial validation setting because of their strong
> complementarity in cardiovascular practice, where they are routinely
> interpreted together.
>
> That said, the present setup primarily validates two-modality synergy and
> does not empirically demonstrate extensibility to more complex
> multi-evidence scenarios. To address this directly, we have conducted
> additional experiments on the publicly available IU X-Ray dataset, which
> differs from the original cardiovascular setting in clinical specialty,
> imaging type, and report format. Specifically, we integrated a new chest
> X-ray image-to-radiology-report specialist agent into HetMedAgent and
> performed an acute/non-acute decision-making task, comparing against existing
> multimodal models. Results are available at https://anonymous.4open.science/r/ICML_rebutal-0B83/1Supplementary%20Scenario.PNG and indicate that the
> framework transfers effectively to this new clinical scenario.
>
> Architecturally, the specialist registry (Equation 3) functions as a
> plug-in interface: adding a new specialist requires only the standardized
> output format (Equation 5), with no changes to the Orchestrator, Reasoning
> Agent, or routing logic. We will make this point more explicit in
> Section 3.2.2, clarify the full input specification, and incorporate the
> IU X-Ray experimental setup, results, and analysis into the revised
> manuscript.
>
> **Q3 -- Subgroup Fairness and Governance.**
> Following this suggestion, we have conducted a disaggregated performance
> analysis across gender (Female/Male) and age (<65, 65–74, 75–84,
> ≥85) subgroups; results are presented in a new figure available at
> https://anonymous.4open.science/r/ICML_rebutal-0B83/2Subgroup%20Analysis.png. Across all three tasks, performance remains largely stable: for
> gender groups, no significant difference is observed in risk
> stratification (AUC $P$=0.146, F1 $P$=0.765) or severity assessment
> (AUC $P$=0.149, F1 $P$=0.158); only etiology classification shows a
> marginal gap in AUC ($P$=0.026), though its F1 difference is
> non-significant ($P$=0.060). Across age groups, no task exhibits a
> significant F1 difference (all $P$>0.12). We acknowledge that the
> Age≥85 subgroup ($n$=47) is small and warrants cautious
> interpretation; we will note this and the absence of comorbidity-level
> analysis as explicit limitations in Section 5. Regarding governance:
> Equation 11 resolves any system-versus-clinician conflict in favor of
> the clinician, and we will add a clarifying sentence in the Ethics
> Statement to make this unambiguous.

---

> > ### Author Rebuttal · Reviewer_zJNm · 2026-04-04
> >
> > I thank the authors for the detailed rebuttal and for providing the supplementary IU X-Ray scenario (Tables S1–S3) and the subgroup fairness figure.
> >
> >     Q1: I thank the authors for the clear statement that momentum smoothing will be described, and that prospective evaluations with actual clinicians accounting for intra-clinician variability and threshold stability are required prior to any deployment claims, with a possible caveat for fragility with noisy feedback if observed.
> >      Q2: The acute versus non-acute IU X-Ray task, patient inter-split, and comparison to ViT-BERT support the idea that the heterogeneous framework can go beyond the first cardiovascular use case.
> >      Q3: Disaggregating gender and age addresses the confusion in the previous rebuttal regarding reported metrics vs. p-values used for comparisons. I highlight that there is a statistically significant gender difference in etiology AUC (p=0.026), but with no differences in F1 and other tasks. I also agree with the addition of the small age ≥ 85 subgroup and lack of comorbidities stratification as additional limitations. Finally, it should be clarified that clinicians' authority is maintained by Equation 11.
> >
> >  Overall, these additions address my concerns in a substantive way; I maintain my positive recommendation and look forward to the revised manuscript.

---

> > > ### Author Response · Authors · 2026-04-04
> > >
> > > We sincerely thank Reviewer zJNm for the thorough and constructive review throughout this process. Your questions on noise robustness, cross-domain generalizability, and subgroup fairness were instrumental in strengthening the paper. We will carefully revise the manuscript to incorporate all discussed improvements and continue to address the remaining concerns raised by other reviewers. We greatly appreciate your time and expertise.

---

### Official Review · Reviewer_M1ez · 2026-03-14

**Soundness:** 3
**Presentation:** 3
**Significance:** 4
**Originality:** 3
**Overall Recommendation:** 5
**Confidence:** 5

**Summary:**

The paper argues that, although LLMs are very powerful, a team of AI agents composed of LLM-based AI agents with specific roles as well as domain-specific agents such as common deep learning-based image diagnosis models, would perform better. For one, it would allow for better uncertainty management with controlled evidence weighting and conflict detection. The approach can work autonomously while requesting human oversight in high-uncertainty cases. The approach shows significant improved performance compared to other medical LLMs and multi-agent systems on clinical decision tasks.

**Compliance With Llm Reviewing Policy:**

Affirmed.

**Final Justification:**

My overall evaluation was already "Accept", so my review was primarily targeted at improving the paper further which worked to a sufficient degree. I maintained my overall evaluation "Accept".

**Key Questions For Authors:**

Q1: Section 3.1 describes that findings have a single confidence score. Does that suffice? For example, instrument precision, confidence in sample integrity, how unique a pattern is, these are all possibly independent facets of uncertainty.
Q2: What is the added value of Algorithm 1. You already did a nice job in formal problem formulation, so you may use this space for further discussions.
Q3: In your experiments, a subset of cases will have been decided upon autonomously and a subset will have had clinical feedback by a human. How does the performance compare between these two? I would expect that the first subset would need to be almost 100% correct.
Q4a: Related to W1, what would it look like, if we would insist on a human clinician to always make the final decision (diagnosis, treatment decision)?
Q4b: Also related to W1, one can of course give to an AI every role it is capable of doing rendering us humans as mere assistants being asked for advice on edge cases. But we can, of course, also decide not to, so from the human perspective, what roles would we humans like to keep for ourselves to retain a fulfilling job as a clinician?
Q5: Related to W2, what human-based team working approaches could be adapted and used to include AI team members in a collaboration much like you present for HetMedAgent?
Q6: What mechanisms would be possible in your approach that would categorically prevent any patient-level information going into a commercial LLM?

**Limitations:**

Limitations are not discussed. Only a bit under "Societal Impact" could be considered as mentioning some limitations.

**Strengths And Weaknesses:**

S1: Although I have some criticisms, I sincerely believe that such a heterogeneous multi-agent approach is the way to go. The reason for this overly extensive review report, is exactly for this reason: I hope my questions and ideas will help and encourage you to further develop the idea into something that would truly improve medical practice, not only for efficiency, quality of clinical decision making, but also for the well-being of the human stakeholders: patients and clinical personnel.
S2: Nice related work section with a diverse set of subsections.
S3: Useful information-rich figures.
S4: Nice diverse set of clinical decision tasks (Table 67 in Appendix B)

W1: "Clinicians themselves are agents", "clinical oversight for high-uncertainty cases", etc. This is a role that humans should play in such an approach indeed, but not only this role. You seem to neglect the principles of responsibility and accountability. In 1979 IBM already said: "A computer can never be held accountable, therefore _a computer must never make a management decision_". You should at least discuss this issue, perhaps as a limitation, because there are ethical concerns with how you have currently designed your approach.
W2: What you describe is an approach for joint decision making in a high-uncertainty context. I suspect that there is decades of literature on the topic of decision making being done by teams of humans. I expected at least a small section in the related work on this topic. But besides that, I would like to advise you to look into this literature more thoroughly to see if there are strong methods that can be adapted to include AI agents in the team-based decision making. Also, look into other domains, for example, criminal investigation where police gathers evidence and judges make the final decisions. Isn't clinical decision making similar with nurses and labs doing the gathering of the evidence and the clinician doing the final diagnosis or treatment decision (see also W1)?
W3: Are you feeding medical information into a commercial LLM? Even pseudonominised, there are privacy risks. This needs discussion on possiblities for how to make it safe. For example, smaller LLMs perform quite nicely on certain tasks; perhaps some of your LLM-based agents could be locally-running smaller LLMs.
W4: In 3.2.7 you define that the decision for asking for human clinical oversight is based on the uncertainty. Wouldn't "risk" be a better criterion, i.e., based on both uncertainty and potential impact? Might also be a relevant discussion point.
W5 (minor): It is presented as a "paradigm shift". I find that a bit too exaggerating. It would mean that the whole world is doing a certain other paradigm and you are the only one who saw the light. It is a good strong idea, so no need to oversell it.
W6 (minor): Figure 6 righthand side is a line chart, but the x-axis doesn't have an order. This is one of the clear "DON'Ts" from data visualization. Turn it into a bar chart or something.
W7: The paper does not have a "Conclusions" section. It just stops with 4.5 the case study.

---

> ### Author Rebuttal · Authors · 2026-03-31
>
> We thank Reviewer M1ez for the positive recognition of our framework and
> the constructive feedback on accountability, team decision-making, and
> privacy.
>
> **W1&Q4a&Q4b -- Accountability and Human Roles.**
> HetMedAgent is designed as a Clinical Decision Support System (CDSS)
> operating under clinician oversight. We will
> revise the Ethics Statement to open with this framing and reference the
> 1979 IBM principle the reviewer mentions. The revised text will enumerate
> the roles that remain with clinicians: issuing final diagnoses and
> treatment orders, managing ambiguous presentations, handling patient
> communication and informed consent, and bearing professional and legal
> accountability. Section 3.2.7 will clarify that even when
> $U(D_{\text{prelim}}) \leq \theta_P$ (Equation 11), the system output is
> a *recommendation*, not an executed order.
>
> **W2&Q5 -- Human Team Decision-Making.**
> We will add a subsection (Section 2.5) situating HetMedAgent within
> Multidisciplinary Team (MDT) workflows such as tumor boards. The
> structural parallel is direct: modality specialists gather evidence while
> an attending physician synthesizes a recommendation, mirroring our
> specialist-agents-plus-orchestrator design. Munyaka et al. (2023) show
> that structured delegation based on individual capability outperforms
> single generalist decision-makers, supporting our heterogeneous agent
> design; Inkpen et al. (2023) demonstrate that joint decision quality
> improves when algorithmic output is matched to user expertise, aligning
> with our uncertainty-based routing (Equation 11). We will also incorporate
> the evidence-collection vs. decision-synthesis analogy the reviewer
> suggests to further motivate this design.
>
> **W3&Q6 -- Data Privacy and Local Deployment.**
> The Orchestrator and Reasoning Agent can be replaced with locally deployed
> open-weights models (e.g., Llama-3-Med, Meditron) without modifying the
> rest of the pipeline, ensuring physical data isolation. We will state this
> in the Impact Statement and recommend local deployment as default for
> HIPAA/GDPR-regulated institutions.
>
> **W4 -- Risk vs. Uncertainty.**
> A case with moderate uncertainty but life-threatening consequences (e.g.,
> missed pulmonary embolism) should still be escalated. We will acknowledge
> the risk–uncertainty distinction in Section 3.2.7 and formalize
> $\text{Risk} = \text{Uncertainty} \times \text{Impact}$ as a future
> direction in Section 5, describing how $\theta_P$ could be modulated by
> task-specific severity scores.
>
> **Q1 -- Multi-Faceted Confidence.**
> Equation 10 decomposes system-level uncertainty into three dimensions
> (confidence, conflict, coherence), but *per-modality*
> facets—instrument precision, sample integrity, pattern
> uniqueness—are not separately captured. Each specialist currently
> outputs a single scalar $c_i$ calibrated via post-hoc temperature
> scaling (Appendix C). Decomposing $c_i$ into finer-grained facets
> requires modality-specific calibration data; we will note this as future
> work.
>
> **Q2 -- Algorithm 1.**
> We will move Algorithm 1 to the appendix and add a concise prose summary
> in Section 3 describing execution order, parallelism, and conditional
> branching (including the clinician feedback loop).
>
> **Q3 -- Autonomous vs. Intervention Performance.**
> Figure 5 (right panel) compares F1 between the two subsets across all
> three tasks at $p < 0.001$ (Mann–Whitney U). Autonomous cases show
> consistently higher F1, confirming that routing correctly identifies
> harder cases for escalation; we will highlight this more prominently.
> The autonomy we emphasize refers to the end-to-end framework
> operation—from evidence collection through conflict-aware fusion to
> routing—rather than replacement of clinician judgment. As noted in W1,
> clinicians retain final decision-making authority.
>
> **W5&W6&W7.**
> In the revised manuscript, "Paradigm shift" will be replaced with "a promising alternative to
> monolithic models." Accordingly, the title will be revised by replacing "Paradigm" with
> "Framework". Figure 6 left panel will be redrawn as a bar chart.
> A Conclusion section (Section 5) will cover limitations and future
> directions.

---

> > ### Author Rebuttal · Reviewer_M1ez · 2026-04-05
> >
> > Thank you for your thorough attention to the concerns and questions I raised. I hope the authors agree with me that the paper improved by addressing these points.

---

> > > ### Author Response · Authors · 2026-04-05
> > >
> > > Thank you sincerely for your kind acknowledgement and for taking the time to carefully evaluate our rebuttal. We are genuinely pleased that the revisions addressed your concerns, and we fully agree that the paper has improved through this process. We will ensure that every commitment made in the rebuttal — including the revised Ethics Statement, the new Section 2.5 on MDT workflows, the updated Figure 6, the revised title, and the new Conclusion section — is faithfully incorporated into the final manuscript.
> > >
> > > Additionally, we would like to mention that we have also submitted a detailed response to Reviewer BJ3A's further questions, which includes additional ablation studies and clarifications on the theoretical formulation. We hope that these supplementary materials may also be of interest to you and contribute to a more comprehensive understanding of our framework.
> > >
> > > Thank you again for your constructive engagement throughout the review process. Your feedback has been invaluable in strengthening this work.

---

### Decision · Program_Chairs · 2026-04-30

**Decision:**

Accept (regular)

**Comment:**

This paper addresses an important and timely question in medical AI: whether future systems should rely on increasingly capable generalist LLMs alone, or instead combine them with modality-specific specialist models and clinician oversight. I find the central thesis compelling, and the proposed HetMedAgent framework provides a concrete and well-executed instantiation of this idea. Across the review process, the paper was generally viewed as technically strong, practically relevant, and well motivated, with particular strengths in its heterogeneous architecture, uncertainty-aware routing, and conflict-aware evidence fusion.

A major strength of the submission is that it goes beyond a purely conceptual proposal and provides a reasonably comprehensive empirical study. The framework is evaluated on three clinical decision-making tasks, with comparisons against both standalone medical LLMs and existing multi-agent baselines, and the paper includes ablations that help justify the contributions of the orchestration and routing components. Reviewers also found the use of specialist models for modality-specific analysis to be a sensible design choice that aligns with how real clinical workflows are often structured.

The main concerns raised during review were about generalizability beyond the initial cardiovascular setting, the realism of the clinician-feedback simulation used in threshold calibration, the interpretation of confidence scores and evidence fusion, and broader ethical/governance issues such as accountability and privacy. In my view, the rebuttal addressed these concerns in a substantive way. In particular, the authors clarified the intended role of the system as a clinical decision support framework rather than an autonomous decision-maker, added discussion of clinician accountability and privacy-preserving local deployment, and strengthened the paper with supplementary validation on an additional IU X-Ray scenario outside the original cardiovascular use case. The additional subgroup analysis and governance clarifications also improve the responsible-ML framing.

I do not think the paper is without limitations. The current evaluation is still centered on a relatively focused set of tasks, the adaptive threshold calibration remains validated using simulated rather than real clinician feedback, and the natural-language interface between specialists and the reasoning agent may lose some modality-specific information. These are real caveats, and they should be clearly discussed in the final version. However, I view them as limitations of scope rather than fatal weaknesses. The core contribution — namely, a heterogeneous multi-agent design pattern that combines generalist reasoning, specialist perception, and uncertainty-aware clinician escalation — remains strong and useful even with these limitations.

Overall, I find this to be a strong paper with meaningful practical relevance and a high level of completeness. The reviewers were broadly positive, and several important concerns were resolved during rebuttal. I therefore recommend Accept.